# Collagenolysis-dependent DDR1 signalling dictates pancreatic cancer outcome

Hua Su[1,10], Fei Yang[2,10], Rao Fu[2], Brittney Trinh[1], Nina Sun[3], Junlai Liu[1], Avi Kumar[4], Jacopo Baglieri[3], Jeremy Siruno[1], Michelle Le[1], Yuhan Li[1], Stephen Dozier[3], Ajay Nair[5], Aveline Filliol[5], Nachanok Sinchai[1], Sara Brin Rosenthal[6], Jennifer Santini[7], Christian M. Metallo[4], Anthony Molina[3], Robert F. Schwabe[5], Andrew M. Lowy[8], David A. Brenner[3], Beicheng Sun[2,9]✉ & Michael Karin[1]✉

Pancreatic ductal adenocarcinoma (PDAC) is a highly desmoplastic, aggressive cancer that frequently progresses and spreads by metastasis to the liver[1]. Cancer-associated fibroblasts, the extracellular matrix and type I collagen (Col I) support[2,3] or restrain the progression of PDAC and may impede blood supply and nutrient availability[4]. The dichotomous role of the stroma in PDAC, and the mechanisms through which it influences patient survival and enables desmoplastic cancers to escape nutrient limitation, remain poorly understood. Here we show that matrix-metalloprotease-cleaved Col I (cCol I) and intact Col I (iCol I) exert opposing effects on PDAC bioenergetics, macropinocytosis, tumour growth and metastasis. Whereas cCol I activates discoidin domain receptor 1 (DDR1)–NF-κB–p62–NRF2 signalling to promote the growth of PDAC, iCol I triggers the degradation of DDR1 and restrains the growth of PDAC. Patients whose tumours are enriched for iCol I and express low levels of DDR1 and NRF2 have improved median survival compared to those whose tumours have high levels of cCol I, DDR1 and NRF2. Inhibition of the DDR1-stimulated expression of NF-κB or mitochondrial biogenesis blocks tumorigenesis in wild-type mice, but not in mice that express MMP-resistant Col I. The diverse effects of the tumour stroma on the growth and metastasis of PDAC and on the survival of patients are mediated through the Col I–DDR1–NF-κB–NRF2 mitochondrial biogenesis pathway, and targeting components of this pathway could provide therapeutic opportunities.

Retrospective clinical studies suggest that patients with PDAC whose tumours have a fibrogenic but inert stroma (defined by extensive extracellular matrix (ECM) deposition, low expression of the myofibroblast marker α-SMA and low levels of matrix metalloprotease (MMP) activity) have improved progression-free survival compared to patients whose tumours are populated by a fibrolytic stroma (defined by a low content of collagen fibres, high expression of α-SMA and high levels of MMP activity)[5]. How the stromal state affects clinical outcome is unknown. Moreover, previous investigations of the influence of the stroma on the growth and progression of PDAC have yielded conflicting results, assigning stroma and cancer-associated fibroblasts (CAFs) as either tumour-supportive[6] or tumour-restrictive[4]. It is likely that the failure of stromal-targeted PDAC therapies[7] is due, in part, to unrecognized pathways that result in tumour-promoting or tumour-suppressive stromal subgroups; successful treatments may thus require precision medicine rather than one-size-fits-all approaches.

## cCol I and iCol I differentially affect PDAC growth

To investigate how the fibrolytic stroma affects PDAC outcome, we compared survival between patients with high and low collagenolysis, using a panel of collagen-cleaving MMPs (MMP1, MMP2, MMP8, MMP9, MMP13 and MMP14), and found that high mRNA expression of MMPs correlated with poor survival (Extended Data Fig. 1a). Single-cell RNA sequencing (scRNA-seq) revealed that *MMP1*, *MMP14* and *MMP2* mRNAs were the most abundant MMP family members, and were expressed in epithelial-tumour cells, M2-like macrophages and fibroblastic cells (Extended Data Fig. 1b). The main target of MMPs in desmoplastic tumours is Col I, the prevalent ECM protein. Using antibodies that distinguish iCol I from cCol I (3/4 Col I; Fig. 1a), we stratified a cohort of 106 patients with PDAC whose tumours had been resected (see below), and correlated the tumour Col I state with survival data. These results also pointed to Col I remodelling as a strong prognostic factor, as patients whose tumours were enriched for cCol I had poorer median survival

[1]Laboratory of Gene Regulation and Signal Transduction, Departments of Pharmacology and Pathology, School of Medicine, University of California San Diego, La Jolla, CA, USA. [2]Department of Hepatobiliary Surgery, Affiliated Drum Tower Hospital of Nanjing University Medical School, Nanjing, China. [3]Department of Medicine, University of California San Diego, La Jolla, CA, USA. [4]Molecular and Cell Biology Laboratory, Salk Institute for Biological Studies, La Jolla, CA, USA. [5]Department of Medicine, Columbia University, New York, NY, USA. [6]Center for Computational Biology and Bioinformatics, Department of Medicine, University of California San Diego, La Jolla, CA, USA. [7]UCSD School of Medicine Microscopy Core, University of California San Diego, La Jolla, CA, USA. [8]Department of Surgery, Division of Surgical Oncology, Moores Cancer Center, University of California San Diego, La Jolla, CA, USA. [9]Department of Hepatobiliary Surgery, First Affiliated Hospital of Anhui Medical University, Hefei, China. [10]These authors contributed equally: Hua Su, Fei Yang. ✉e-mail: sunbc@nju.edu.cn; karinoffice@health.ucsd.edu

(Fig. 1b). To understand the basis for these results and mimic a cCol I<sup>low</sup> inert tumour stroma, we used mice expressing either wild-type *Col1a1*[+/+] (Col I[WT]), or a MMP-resistant version of Col I generated by two amino acid substitutions in the 1α1 subunit that block the cleavage of Col I by MMPs[8], *Col1a1*[r/r] (Col I[r/r]). Col I[r/r] mice develop more-extensive hepatic fibrosis than Col I[WT] mice, but despite the hepatocellular carcinoma (HCC)-supportive functions of hepatic fibrosis[9], they poorly accommodate HCC growth, through unknown mechanisms[10]. Col I[WT] and Col I[r/r] mice were either orthotopically or intrasplenically (to model liver metastasis) transplanted with mouse PDAC KPC960 (KPC) or KC6141 (KC) cells. Col I[r/r] mice poorly supported the growth of primary pancreatic tumours or hepatic metastases, even though their pancreata were more fibrotic than Col I[WT] pancreata. These differences persisted in mice that were pretreated with the pancreatitis inducer caerulein (CAE), which stimulated liver metastasis in Col I[WT] pancreata (Fig. 1c,d and Extended Data Fig. 1c–f). After intrasplenic transplantation, KPC or KC tumours in Col I[WT] livers were larger in mice pretreated with CCl₄ to induce liver fibrosis, whereas the number and size of tumours were lower in Col I[r/r] livers, regardless of CCl₄ pretreatment (Fig. 1e,f and Extended Data Fig. 1g). As expected, Col I[r/r] livers were more fibrotic than Col I[WT] livers, regardless of CCl₄ pretreatment (Extended Data Fig. 1h). Primary PDAC and liver metastases were confirmed by staining with ductal (CK19), progenitor (SOX9) or proliferation (Ki67) markers (Extended Data Fig. 1e,f,i). Enhanced tumour growth in CAE- or CCl₄-pretreated Col I[WT] mice suggested that tumour suppression in Col I[r/r] mice was not simply due to a space limitation imposed by a build-up of Col I. To determine how Col I remodelling affects human PDAC, we subcutaneously co-transplanted wild-type and R/R fibroblasts with a patient-derived xenograft cell line (1305) into immunocompromised *Nu/Nu* mice. Wild-type fibroblasts enhanced tumour growth, whereas R/R fibroblasts inhibited tumour growth but lost their inhibitory activity after ablation of *Col1a1* (Fig. 1g) whose loss did not affect the stimulatory activity of wild-type fibroblasts, suggesting a specific inhibitory function of noncleaved Col I.

## The Col I state controls PDAC metabolism

To determine the basis for reduced tumorigenesis in Col I[r/r] mice, we plated KPC cells on ECM deposited by wild-type and R/R fibroblasts, incubated them in low-glucose (LG) medium (to model nutrient restriction) and performed RNA sequencing (RNA-seq). Bioinformatic analysis revealed marked differences between cells cultured on wild-type and cells cultured on R/R ECM, with the former showing an upregulation of signatures related to sulfur amino acid metabolism, mammary gland morphogenesis, telomere maintenance and RNA processing, and the latter showing an upregulation of mRNAs related to innate immunity and inflammation (Extended Data Fig. 2a). The most notable differences were in nuclear and mitochondrial genes that encode components of the mitochondrial electron transfer chain (ETC) and ribosome subunits, and macropinocytosis-related genes, which were upregulated by wild-type and suppressed by R/R ECM (Fig. 2a–c). Consistent with the upregulation of macropinocytosis-related genes by wild-type ECM, IKKα-deficient KC cells, which have high macropinocytosis activity[11], grew better than parental cells in Col I[WT] livers, but grew as poorly as parental KC cells in Col I[r/r] livers (Extended Data Fig. 1g).

To assess the effects of Col I on metabolism, we labelled wild-type and R/R fibroblasts with [³H]-proline or [U-¹³C]-glutamine for five days, during which period the cells coated the plates with Col I-containing ECM. After decellularization, KPC or KC cells and variants thereof were plated and cultured for 24 h in LG medium. The uptake of [³H] in cells plated on wild-type ECM was dependent on macropinocytosis, as indicated by sensitivity to macropinocytosis inhibitors (EIPA (an NHE1 inhibitor), IPI549 (a PI3Kγ inhibitor) or MBQ-167 (a CDC42 and RAC inhibitor)) and to the knockdown of NHE1 or SDC1, and enhancement by the ULK1 inhibitor MRT68921 (MRT)[11]. By contrast, cells plated on R/R ECM showed a negligible uptake of [³H] that was unaffected by the inhibition

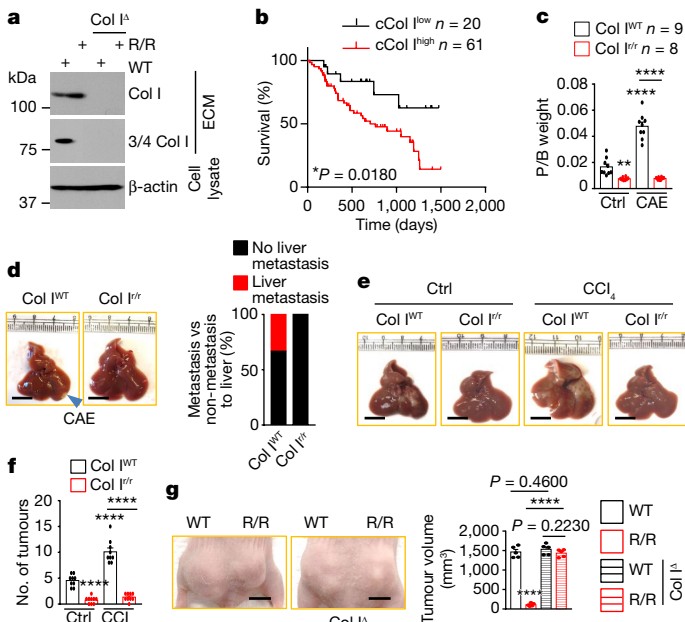

**Fig. 1 | Col I cleavage controls PDAC growth. a**, Immunoblot showing the specificity of antibodies to iCol I and cCol I (3/4 Col I) in ECM produced by the indicated fibroblasts. Col I[Δ], Col I knockout; WT, wild type. **b**, Overall survival of patients with resected PDAC stratified according to cCol I expression (shown in Fig. 5a). Significance was determined by log-rank test. **c**, Pancreas weight relative to body weight (P/B weight) four weeks after orthotopic KPC cell transplantation into Col I[WT] or Col I[r/r] mice that were pretreated with CAE or without CAE. Ctrl, control. **d**, Liver morphology in CAE-treated mice. Liver metastases were detected in 33% of Col I[WT] mice. **e,f**, Liver gross morphology (**e**) and tumour numbers (**f**) two weeks after intrasplenic transplantation of KPC cells into Col I[WT] or Col I[r/r] mice with or without CCl₄ pretreatment. **g**, Representative images and sizes of subcutaneous tumours formed by human 1305 cells co-transplanted with WT, R/R or Col I[Δ] WT or R/R fibroblasts into *Nu/Nu* mice. Data in **f** (*n* = 9 mice), **g** (*n* = 5 mice) and **c** are mean ± s.e.m. Statistical significance determined by two-tailed *t*-test. Exact *P* values in **c,f** are shown in the Source Data. ****P < 0.0001. Scale bars (**d,e,g**), 1 cm.

of macropinocytosis (Extended Data Fig. 2b–e). Notably, ablation of *Col1a1* or overexpression of cleavable Col I in ECM-laying R/R fibroblasts restored [³H] uptake (Extended Data Fig. 2b). Cells that were cultured on ¹³C-glutamine-labelled wild-type ECM took up glutamine and metabolized it, but cells that were plated on ¹³C-glutamine-labelled R/R ECM exhibited minimal glutamine uptake and metabolism (Fig. 2d,e). Congruently, cells that were cultured on wild-type ECM had higher levels of ATP and a higher amino acid content than cells that were cultured on R/R ECM, and this effect was further increased by treatment with MRT and reduced by blockade of macropinocytosis; by contrast, cells that were cultured on R/R ECM had low levels of ATP and amino acids, which were barely affected by the inhibition of macropinocytosis (Fig. 2f and Extended Data Fig. 2f–j). Ablation of Col I or overexpression of wild-type Col I prevented the decline in ATP and amino acids (Extended Data Fig. 2h,j), suggesting that cCol I is a key signalling molecule that stimulates PDAC metabolism and energy generation.

## cCol I to iCol I ratio controls DDR1–NRF2 signalling

KPC or human MIA PaCa-2 cells plated on wild-type ECM or co-cultured with wild-type fibroblasts in LG or low-glutamine (LQ) medium exhibited high rates of macropinocytosis, as measured by their uptake of tetramethylrhodamine-labelled high-molecular-mass dextran (TMR-DEX), whereas cells plated on R/R ECM or co-cultured with R/R fibroblasts exhibited low rates of macropinocytosis (Fig. 3a and

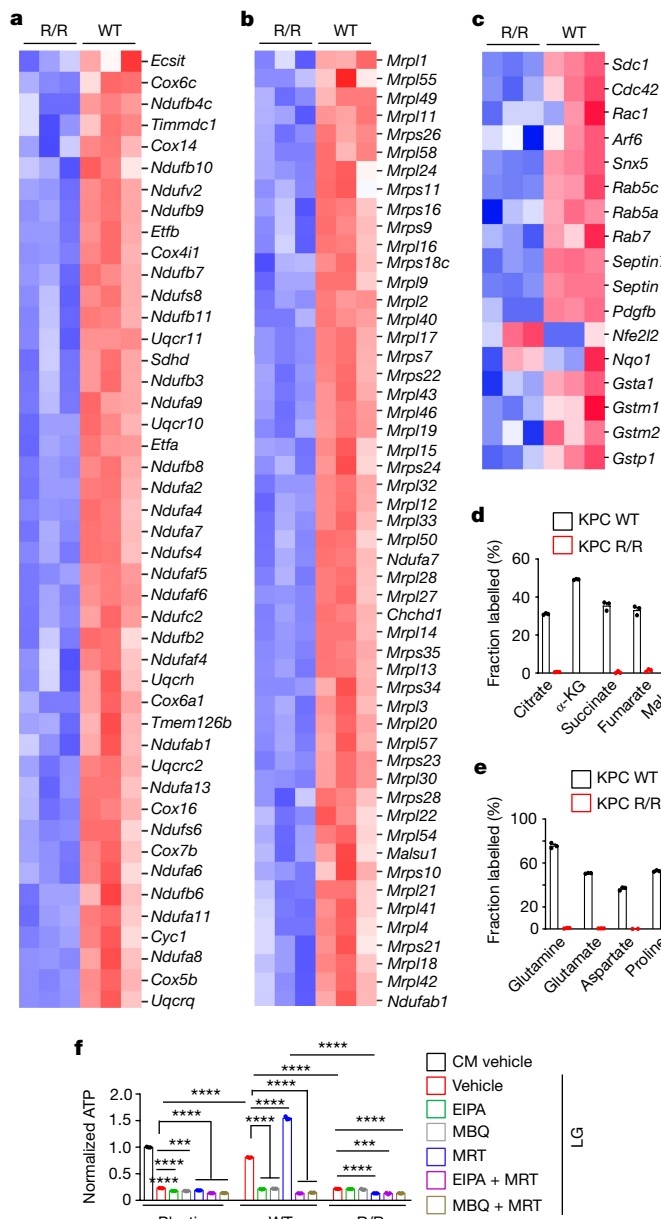

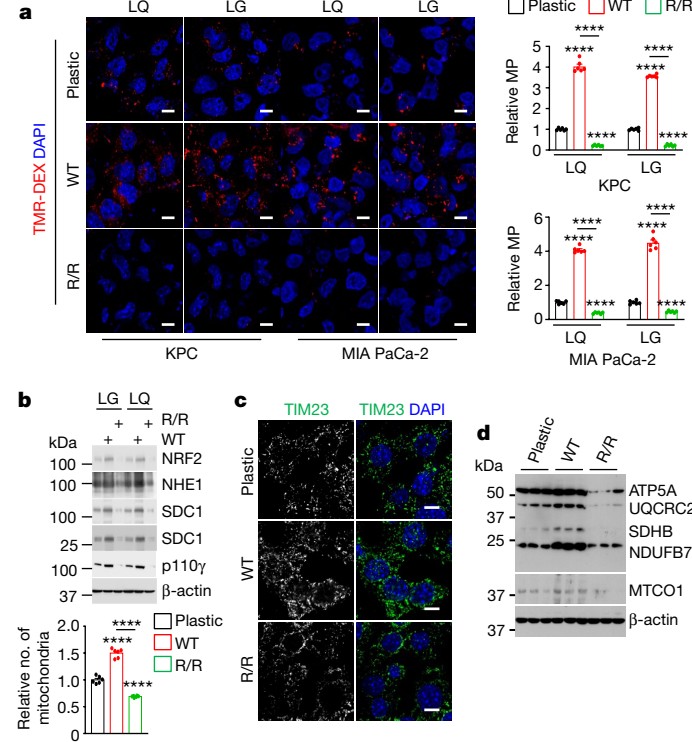

**Fig. 2 | Col I cleavage controls PDAC metabolism. a–c,** Genes differentially expressed between KPC cells grown on wild-type or R/R ECM in LG (0.5 mM) medium for 24 h. Blue, replicates with low expression (z-score = −2); red, replicates with high expression (z-score = 2). Mitochondrial ETC genes (**a**), mitochondrial ribosome subunit genes (**b**) and macropinocytosis-related and NRF2-target genes (**c**). **d,e,** Fractional labelling (mole per cent enrichment) of TCA cycle intermediates (**d**) and intracellular amino acids (**e**) in KPC cells incubated for 24 h in LG medium after plating on [U-$^{13}$C]-glutamine-labelled wild-type or R/R ECM. α-KG-, α-ketoglutarate. **f,** KPC cells plated on wild-type or R/R ECM or plastic were incubated in CM or LG medium with or without EIPA, MBQ-167 (MBQ), MRT68921 (MRT), EIPA + MRT or MBQ + MRT for 24 h. Total cellular ATP is presented relative to untreated plastic-plated cells. CM, complete medium. Data in **d,e** (n = 3 per condition) and **f** (n = 3 independent experiments) are mean ± s.e.m. Statistical significance determined by two-tailed t-test. Exact P values are shown in the Source Data. ***P < 0.001; ****P < 0.0001.

Extended Data Fig. 3a). Furthermore, KPC cells cultured on wild-type ECM showed a marked upregulation of macropinocytosis-related proteins and NRF2 relative to plastic-cultured cells, but culturing on R/R ECM had the opposite effect (Fig. 3b). Similar differences in macropinocytosis activity, NRF2 and macropinocytosis-related mRNAs and proteins were shown by KPC tumours in Col I^WT or Col I^r/r pancreata or

**Fig. 3 | Col I cleavage controls macropinocytosis and the number of mitochondria in PDAC. a,** Representative images and rates of macropinocytosis (MP) in TMR-DEX-incubated KPC and MIA PaCa-2 cells grown on plates with or without wild-type or R/R ECM and incubated in LQ or LG medium for 24 h. **b,** Immunoblot analysis of the indicated proteins in KPC cells treated as in **a**. **c,** Representative images of mitochondria (TIM23) in KPC cells grown on plates with or without wild-type or R/R ECM and incubated in LG medium for 24 h. Bottom left, quantification of the number of mitochondria. **d,** Immunoblot analysis of the indicated proteins in KPC cells treated as in **c**. Results in **a,c** (n = 6 fields) are mean ± s.e.m. Statistical significance determined by two-tailed t-test. ****P < 0.0001. Scale bars (**a,c**), 10 μm.

livers (Extended Data Fig. 3b–d). Mitochondria are important for cancer growth in that they generate energy for macromolecular synthesis[12]. Consistent with the RNA-seq data, mitochondria and ETC proteins were decreased in PDAC cells grown on R/R ECM or in Col I^r/r pancreata (Fig. 3c,d and Extended Data Fig. 3e).

The human PDAC stroma consists of intact and cleaved collagens. To recapitulate this setting and determine how the balance of iCol I to cCol I affects PDAC metabolism, we mixed R/R fibroblasts with wild-type (R:W) or Col I^Δ (knockout) (R:KO) fibroblasts to generate ECM with different amounts of iCol I and cCol I, and confirmed this with isoform-specific antibodies. KPC cells were plated on the ECM preparations and kept in LG medium for 24 h, and their rates of macropinocytosis, numbers of mitochondria and levels of nuclear NRF2 were evaluated. Nondegradable Col I at 6:4 (R:W) or 4:6 (R:KO) ratios and higher ratios inhibited macropinocytosis and reduced mitochondria numbers and nuclear NRF2 (Extended Data Fig. 3f,g). We conclude that iCol I inhibits macropinocytosis and mitochondrial biogenesis, which are stimulated by different cleaved collagens, not just cCol I.

To investigate how Col I regulates macropinocytosis and mitochondrial biogenesis, we systematically ablated (Extended Data Fig. 4a) all known collagen receptors expressed by KPC cells—MRC2, DDR1, LAIR1 and β1 integrin (ITGB1). The only receptor whose ablation inhibited macropinocytosis activity and mitochondrial biogenesis (Fig. 4a) was DDR1, a collagen-activated receptor tyrosine kinase (RTK)[13], which scRNA-seq showed was highly expressed in primary and liver-metastatic human PDAC epithelial-tumour cells, marked by the mRNA expression

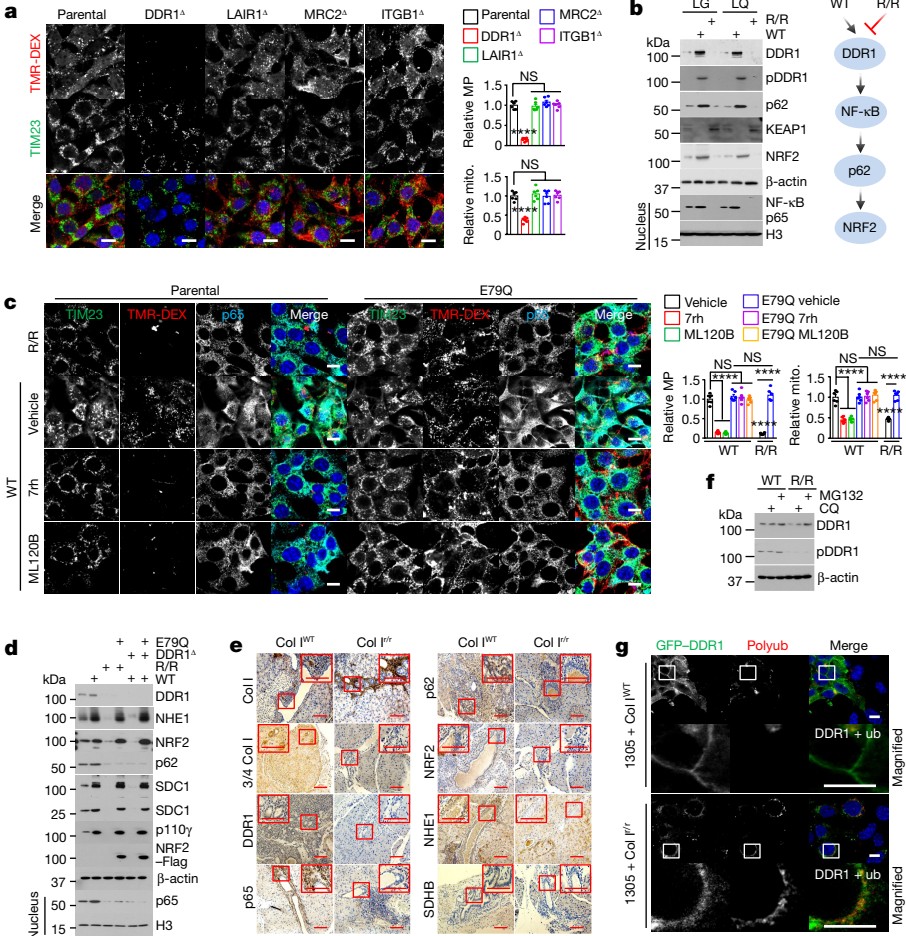

**Fig. 4 | The Col I–DDR1–NRF2 axis controls macropinocytosis and mitochondrial biogenesis. a**, Representative images and quantification of mitochondria and macropinocytosis in TMR-DEX-incubated parental and variant KPC cells grown on wild-type ECM. **b**, Immunoblot analysis of the indicated proteins in KPC cells grown on plastic or wild-type or R/R ECM and incubated in LG or LQ medium for 24 h. The effects of wild-type and R/R ECM on DDR1 signalling are summarized on the right. mito., mitochondria; pDDR1, phosphorylated DDR1. **c**, Representative images and quantification of mitochondria and macropinocytosis in TMR-DEX-incubated parental and NRF2$^{E79Q}$ (E79Q) KPC cells plated on wild-type or R/R ECM in LG medium with or without 7rh or ML120B for 24 h. **d**, Immunoblot analysis of the indicated proteins in parental, E79Q, DDR1$^Δ$ and E79Q/DDR1$^Δ$ KPC cells plated with

or without wild-type or R/R ECM and incubated in LG medium for 24 h. **e**, Representative IHC of the indicated proteins in Col I$^{WT}$ and Col I$^{r/r}$ pancreata four weeks after KPC cell transplantation. Boxed areas are further magnified. Scale bars, 100 µm. **f**, Immunoblot analysis of the indicated proteins in KPC cells plated on wild-type or R/R ECM and incubated in LG medium with or without +MG132 or chloroquine (CQ) for 24 h. **g**, Representative images showing GFP–DDR1 and polyubiquitin (polyub) colocalization in GFP–DDR1-expressing 1305 cells co-cultured with wild-type or R/R fibroblasts in LG medium for 24 h. Boxed areas are further magnified. Data in **a,c** (n = 6 fields) are mean ± s.e.m. Statistical significance determined by two-tailed t-test. Exact P values are shown in the Source Data. ****P < 0.0001; NS, not significant. Scale bars (**a,c,g**), 10 µm.

of *EPCAM* and *KRT19* (Extended Data Fig. 4b). Other collagen receptor mRNAs were either not expressed in PDAC (*LAIR1* and *MRC2*) or had a broad distribution (*ITGB1*). Whereas wild-type ECM stimulated the expression and phosphorylation of DDR1, R/R ECM strongly downregulated DDR1 and its downstream effector NF-κB[14], as well as p62 (Fig. 4b), an NF-κB target[15]. The inhibitory effect of iCol I was not observed in previous DDR1 signalling studies, which used artificially fragmented acid-solubilized collagens as ligands[16]. Consistent with the induction of p62, wild-type ECM decreased KEAP1 and upregulated NRF2, whereas R/R collagen had the opposite effect (Fig. 4b). We wondered whether cCol I affects macropinocytosis and mitochondrial biogenesis through the DDR1–NF-κB–p62–NRF2 cascade. Indeed, R/R ECM and inhibition or ablation of NRF2, DDR1 or IKKβ decreased macropinocytosis activity, 3/4 Col I fragment uptake, NRF2 nuclear localization, mitochondria number and expression of macropinocytosis-related and mitochondrial ETC proteins (Fig. 4c,d and Extended Data Figs. 4c–g and 5a–e). Overexpression of an activated NRF2(E79Q) variant reversed the inhibitory effects of R/R ECM, DDR1 inhibition or IKKβ inhibition

but did not restore or affect DDR1 expression or phosphorylation and p65 nuclear localization. Consistent with these data, pancreatic and liver tumours from Col I$^{r/r}$ mice showed more-extensive expression of iCol I but no cCol I and lower levels of DDR1, p65, p62, NRF2, NHE1 and SDHB (a mitochondrial marker), as compared to tumours from Col I$^{WT}$ mice (Fig. 4e and Extended Data Fig. 5f,g). These results suggest that Col I controls macropinocytosis and mitochondrial biogenesis through the DDR1–NF-κB–p62–NRF2 axis. As myofibroblast-specific ablation of Col I enhances intrahepatic PDAC growth[17], we examined how Col I$^Δ$ ECM affects macropinocytosis and DDR1 signalling. Notably, Col I$^Δ$ ECM behaved like wild-type ECM, stimulating macropinocytosis, mitochondrial biogenesis and DDR1 phosphorylation, which were blocked by the ablation of DDR1 (Extended Data Fig. 6a–c). However, collagen-free ECM generated by Col I$^Δ$ fibroblasts and treatment with bacterial collagenase no longer activated DDR1 and its downstream effectors (Extended Data Fig. 6d). These results are consistent with DDR1 being a general collagen receptor[13], with other collagens in Col I$^Δ$ fibroblasts acting as ligands.

### iCol I triggers DDR1 proteasomal degradation

The expression and function of DDR1 vary in different cancer stages and types[18–21]. Levels of mouse *Ddr1* mRNA were increased by culturing KPC cells on R/R ECM (Extended Data Fig. 6e), implying that the diminished expression of DDR1 protein in these cultures is post-transcriptional. Indeed, MG132, a proteasome inhibitor, but not the lysosomal inhibitor chloroquine, rescued DDR1 expression but not autophosphorylation (Fig. 4f). Notably, GFP–DDR1 showed cell-surface localization and little polyubiquitin colocalization in human 1305 cells that were co-cultured with wild-type fibroblasts, but was cytoplasmic and colocalized with polyubiquitin in R/R fibroblast cocultures (Fig. 4g). Unlike DDR1 in triple-negative breast cancer (TNBC)[20], no shedding of the DDR1 extracellular domain was detected (Extended Data Fig. 6f). Our results therefore reveal a new mode of DDR1 regulation in PDAC and probably in other desmoplastic cancers.

### NRF2 controls mitochondrial biogenesis

ECM from fibroblasts treated with the FDA-approved MMP inhibitor Ilomastat behaved like R/R ECM (Extended Data Fig. 6g,h), indicating that the results were not unique to the Col I[R] variant. R/R ECM also decreased the number of mitochondria in autophagy-deficient PDAC cells (Extended Data Fig. 6i), which suggests that the reduced mitochondrial content is not mediated by mitophagy. Moreover, colocalization of mitochondria and polyubiquitin, which marks mitophagy, was rarely observed (Extended Data Fig. 6j). Expression of TFAM, a key activator of mitochondrial DNA transcription, replication and biogenesis[22], was downregulated in PDAC cells cultured in R/R ECM, but *Nrf1* (unrelated to NRF2) mRNA, PGC1α protein and AMPK activity, which also stimulate mitochondrial biogenesis[23], were upregulated (Extended Data Fig. 6e,k). The latter results match the low ATP content of R/R-ECM-cultured cells. In silico analysis revealed putative NRF2-binding sites in the *Tfam* promoter region, to which NRF2 was recruited in cells plated on wild-type ECM or in NRF2(E79Q)-expressing cells (Extended Data Fig. 6l,m), confirming that NRF2 mediates cCol I-stimulated macropinocytosis and mitochondrial biogenesis.

### Higher levels of iCol I correlate with improved survival

Immunohistochemistry (IHC) of surgically resected human PDAC showed that most tumours (77/106) contained high amounts of 3/4 Col I and most of them exhibited higher levels of staining for DDR1 (58/77), NF-κB p65 (55/77), NRF2 (60/77), SDC1 (53/77), CDC42 (52/77), SDHB (62/77), α-SMA (56/77) and MMP1 (52/77) than did cCol I[low] tumours (Fig. 5a and Extended Data Fig. 7a,b), suggesting that PDAC tumours with fibrolytic stroma have higher macropinocytosis activity and mitochondrial content than do tumours with inert stroma. Moreover, DDR1 and p65, DDR1 and NRF2, p65 and NRF2, NRF2 and macropinocytosis proteins (NHE1, SDC1 or CDC42), and NRF2 and SDHB showed strong positive correlations (Extended Data Fig. 7b), suggesting that the fibrolytic stroma stimulates macropinocytosis and mitochondrial biogenesis through the DDR1–NF-κB–NRF2 axis in human PDAC. Increased levels of cCol I also correlated with high expression of inflammatory markers (Extended Data Fig. 7c), supporting the notion that inflammation may drive Col I remodelling. Notably, patients with cCol I[high] and DDR1[high], cCol I[high] and NRF2[high] or DDR1[high] and NRF2[high] tumours had a considerably worse median survival than did patients with low expression of these markers (Fig. 5b). These results are consistent with those obtained in our preclinical PDAC models, suggesting that the fibrolytic stroma may drive the recurrence of human PDAC through NRF2-mediated macropinocytosis and mitochondrial biogenesis.

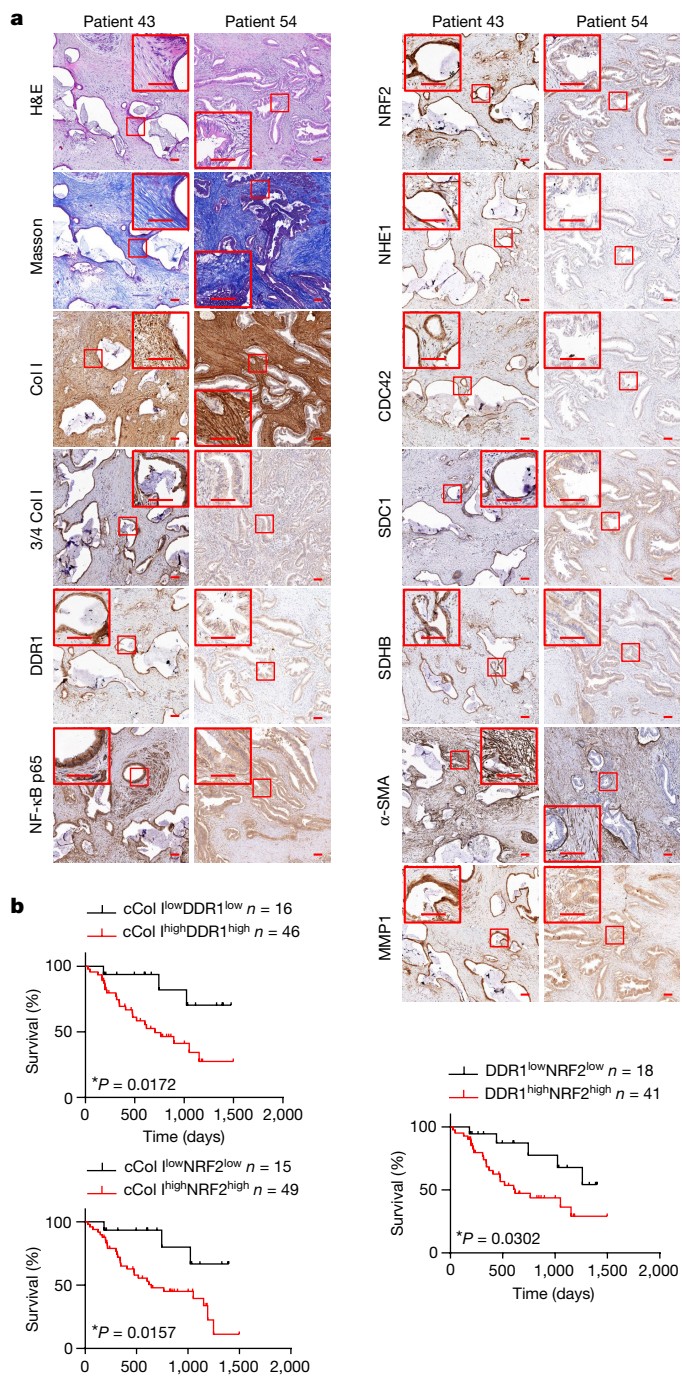

**Fig. 5 | Col I cleavage and increased DDR1–NRF2 signalling predict poor patient survival. a**, Representative IHC of 106 resected human PDAC tissues. H&E, haematoxylin and eosin. Boxed areas are further magnified. Scale bars, 100 μm. **b**, Comparisons of overall survival between patients stratified according to cCol I, DDR1 and NRF2 expression. Significance was determined by log-rank test.

### Targeting the DDR1–NF-κB–NRF2 cascade

Increasing iCol I in the ECM inhibited cellular DNA synthesis (Extended Data Fig. 8a). Parental, NRF2[E79Q] or IKKα-knockdown (IKKα[KD]) PDAC cells were plated on wild-type or R/R ECM, incubated in LG medium and treated with inhibitors of DDR1 (7rh), IKKβ (ML120B), NRF2 (ML385) or macropinocytosis (NHE1[KD] or EIPA, IPI549 or MBQ-167). Whereas wild-type ECM increased and R/R ECM decreased parental PDAC cell growth, inhibition of macropinocytosis, DDR1, IKKβ or NRF2 decreased growth on wild-type ECM (Fig. 6a and Extended Data

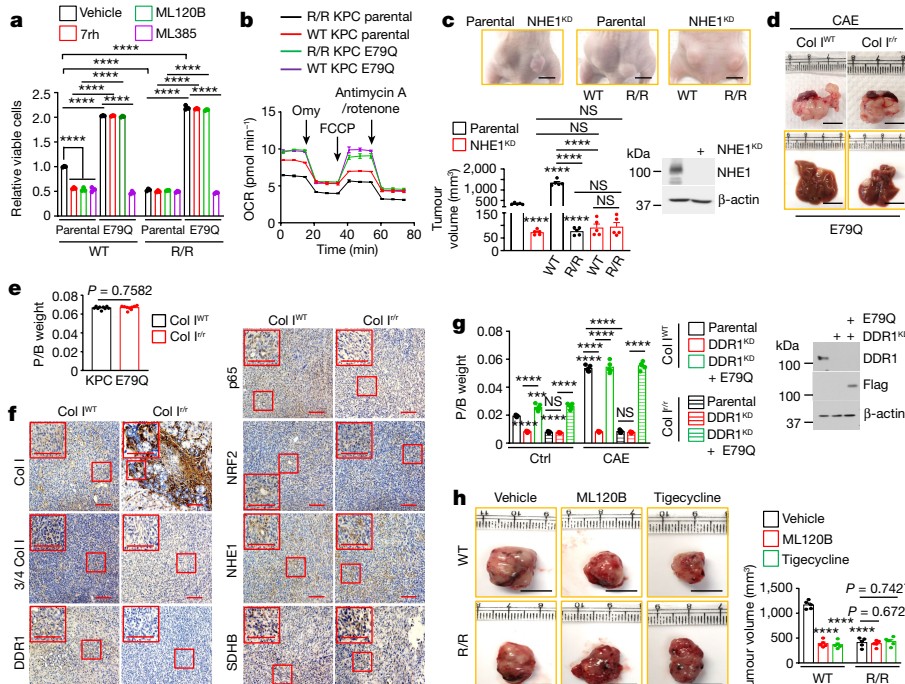

**Fig. 6 | Therapeutic targeting of the DDR1–NF-κB–NRF2 axis inhibits PDAC growth and metabolism. a**, Parental and E79Q KPC cells plated on wild-type or R/R ECM were incubated in LG medium with or without 7rh, ML120B or ML385. Total viable cells are presented relative to parental cells that were treated with vehicle and plated on wild-type ECM. **b**, Oxygen consumption rate (OCR) of parental and E79Q KPC cells plated on wild-type or R/R ECM and incubated in LG medium for 24 h before and after treatment with oligomycin (Omy), FCCP or rotenone/antimycin A. **c**, Representative images and sizes of parental and NHE1$^{KD}$ MIA tumours grown with or without wild-type or R/R fibroblasts in nude mice. Right, immunoblot analysis of NHE1 in MIA cells. **d,e**, Liver and pancreas morphology (**d**) and weight (**e**) four weeks after orthotopic transplantation of KPC E79Q cells into CAE-pretreated Col I$^{WT}$ and Col I$^{r/r}$ mice.

**f**, IHC of pancreatic sections from the mice in **d,e**. Boxed areas are further magnified. Scale bars, 100 μm. **g**, P/B weight four weeks after orthotopic transplantation of the indicated KPC cells into Col I$^{WT}$ and Col I$^{r/r}$ mice pretreated with or without CAE. Right, immunoblot analysis of DDR1 and Flag-tagged E79Q in the indicated KPC cells plated on wild-type ECM in LG medium for 24 h. **h**, Representative images and sizes of MIA tumours grown with wild-type or R/R fibroblasts in nude mice with or without ML120B or tigecycline. Data in **a** (n = 3 independent experiments), **c,g,h** (n = 5 mice) and **e** (n = 9 mice) are mean ± s.e.m. Statistical significance determined by two-tailed t-test. ***P < 0.001, ****P < 0.0001; NS, not significant. Exact P values in **a,c,g** are shown in the Source Data. Scale bars (**c,d,h**), 1 cm.

Fig. 8b–f). NRF2(E79Q)-expressing cells grew faster than parental cells and were resistant to R/R ECM, DDR1 inhibition or IKKβ inhibition but not NRF2 inhibition. IKKα$^{KD}$ cells with high rates of macropinocytosis and high levels of nuclear NRF2 also grew faster than parental cells on wild-type ECM but were more sensitive to R/R ECM and macropinocytosis inhibitors (Extended Data Fig. 8b,c). Inhibition of macropinocytosis, DDR1, IKKβ or NRF2 did not decrease the low growth of parental cells on R/R ECM (Fig. 6a and Extended Data Fig. 8b–f). Moreover, parental KPC or 1305 cells that were plated on wild-type ECM were more sensitive to the mitochondrial protein synthesis inhibitor tigecycline than cells plated on R/R ECM or DDR1$^{KD}$ cells grown on wild-type ECM (Extended Data Fig. 8g). NRF2$^{E79Q}$ cells showed higher rates of oxygen consumption and mitochondrial ATP production than did parental cells; these rates were diminished by R/R ECM but only in the parental cells (Fig. 6b and Extended Data Fig. 8h). Thus, the fibrolytic stroma may support PDAC cell growth through Col I-stimulated macropinocytosis and mitochondrial biogenesis. R/R fibroblasts inhibited human PDAC (MIA PaCa-2) tumour growth, but wild-type fibroblasts were stimulatory. NHE1 ablation or EIPA inhibited tumour growth with or without co-transplanted wild-type fibroblasts or in wild-type livers, but had little effect on tumour growing with R/R fibroblasts or in Col I$^{r/r}$ livers (Fig. 6c and Extended Data Fig. 8i). Tumours growing with wild-type fibroblasts were more fibrotic than tumours without added fibroblasts, and small tumours growing with R/R fibroblasts had the highest collagen content (Extended Data Fig. 8j), indicating that deposition of Col I enhances the growth of PDAC only when Col I is cleaved by MMPs. NRF2$^{E79Q}$ cells in Col I$^{r/r}$ hosts exhibited similar growth, NRF2,

NHE1 and SDHB expression and liver metastases to cells growing in Col I$^{WT}$ hosts, despite low expression of DDR1 and p65 (Fig. 6d–f and Extended Data Fig. 9a).

In TNBC, DDR1 aligns collagen fibres to exclude immune cells[20]. By measuring second-harmonic generation (SHG), we observed no change in collagen fibre alignment and CD8$^+$ T cell content between tumours from Col I$^{WT}$ and Col I$^{r/r}$ pancreata or between parental and DDR1$^{KD}$ tumours, although CD45-, F4/80- or CD4-expressing cells were reduced in tumours from Col I$^{r/r}$ pancreata (Extended Data Fig. 9b,c). Accordingly, ablation of DDR1 inhibited tumour growth, p65, p62, NRF2, NHE1 and SDHB expression in Col I$^{WT}$ pancreata but did not reduce it further in Col I$^{r/r}$ pancreata (Fig. 6g and Extended Data Figs. 9d,e and 10a). NRF2(E79Q) rescued tumour growth and the expression of NHE1 and SDHB—but not p65 or p62—in DDR1$^{KD}$ cells, regardless of Col I status. Similar results were observed in immunodeficient mice (Extended Data Fig. 10b), indicating that the effects of Col I–DDR1 interaction differ between PDAC and TNBC. Notably, inhibition of IKKβ, mitochondrial protein synthesis, TFAM or NRF2 decreased the growth of tumours that were co-transplanted with wild-type fibroblasts or grown in Col I$^{WT}$ pancreata, but had no effect on tumours that were co-transplanted with R/R fibroblasts or grown in Col I$^{r/r}$ pancreata (Fig. 6h and Extended Data Fig. 10c,d), illustrating different ways of targeting PDAC with fibrolytic stroma.

## Discussion

We show here that Col I remodelling is a prognostic indicator for the survival of patients with PDAC. In preclinical models, Col I remodelling

modulated tumour growth and metabolism through a DDR1–NF-κB–p62–NRF2 cascade that is activated by cCol I and inhibited by iCol I. The activation of DDR1 by collagens and downstream activation of NF-κB have been described before[14,16]. However, it was previously unknown—to our knowledge—that iCol I triggers the polyubiquitylation and proteasomal degradation of DDR1. This indicates that DDR1 distinguishes cleaved from intact collagens, and that the latter are capable of restraining the metabolism and growth of tumours. Although inhibition of DDR1 reduces the growth of mouse PDAC[24], the ability of DDR1 to control tumour metabolism by stimulating macropinocytosis and mitochondrial biogenesis was unknown. It is unclear, however, why DDR1—a rather weak RTK[13]—exerts such profound metabolic effects on PDAC cells that express more potent RTKs, such as EGFR and MET. Perhaps this is due to high concentrations of cCol I in the PDAC tumour microenvironment and the stronger NF-κB-activating capacity of DDR1 relative to other RTKs. Indeed, IKKβ inhibition was as effective as the blockade of mitochondrial protein synthesis in curtailing the growth of PDAC with fibrolytic stroma. The differential effects of fibrolytic and inert tumour stroma on PDAC growth and metabolism explain much of the controversy that surrounds the effects of CAFs and Col I on the progression of PDAC in mice[6,17]. Most notably, our findings extend to humans and suggest that Col I remodelling is linked to tumour inflammation. We thus propose that treatments that target DDR1–IKKβ–NF-κB–NRF2 signalling and mitochondrial biogenesis should be evaluated in prospective clinical trials that include stromal state—an important modifier of tumour growth—as an integral biomarker. Given that three Col I-cleaving MMPs were highly expressed in the human PDAC samples we analysed, and that this situation may differ from patient to patient[25], specific MMP inhibitors are additional candidates for precision therapy. A deeper understanding of whether stromal state is affected by neoadjuvant chemotherapy and how it affects metastasis is another area of priority for further investigation. Although our results do not apply to TNBC, they provide mechanistic insight into SPARC-mediated PDAC progression[26,27], and may be applicable to other desmoplastic and fibrolytic cancers.

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

# Methods

## Cell culture

All cells were incubated at 37 °C in a humidified chamber with 5% $CO_2$. MIA PaCa-2 (MIA), UN-KPC-960 (KPC) and UN-KC-6141 (KC) cells, wild-type and R/R fibroblasts were maintained in Dulbecco's modified Eagle's medium (DMEM) (Invitrogen) supplemented with 10% fetal bovine serum (FBS) (Gibco). MIA cells were purchased from ATCC. KPC and KC cells were generated at the laboratory of S. K. Batra[28]. Wild-type and R/R fibroblasts were generated at the laboratory of D.B.[10]. The 1305 primary human PDAC cells were generated by the A.M.L. laboratory from a human PDAC patient-derived xenograft[11] and were maintained in RPMI (Gibco) supplemented with 20% FBS and 1 mM sodium pyruvate (Corning). All media were supplemented with penicillin (100 mg ml$^{-1}$) and streptomycin (100 mg ml$^{-1}$). All cells were partially authenticated by visual morphology. Wild-type and R/R fibroblasts were partially authenticated by ECM production and collagen type I alpha 1 cleavage. KPC and KC cells were partially authenticated by orthotopic tumour formation in mouse pancreas. MIA and 1305 cells were partially authenticated by subcutaneous tumour formation in nude mice. Cells were not further authenticated. Cell lines were tested for mycoplasma contamination. LG medium: glucose-free DMEM medium was supplemented with 0.5 mM glucose in the presence of 10% dialysed FBS and 25 mM HEPES. LQ medium: glutamine-free DMEM medium was supplemented with 0.2 mM glutamine in the presence of 10% dialysed FBS and 25 mM HEPES.

## Plasmids

For gene ablations, the target cDNA sequences (Supplementary Table 1) of mouse *Ddr1*, *Mrc2*, *Itgb1*, *Lair1*, *Nrf2*, *Col1a1* and human *DDR1* were cloned into a lentiCRISPR v2-Blast vector or lentiCRISPR v2-puro vector, respectively using BsmBI. For gene knockdowns, pLKO.1-puro-Ddr1 (TRCN0000023369), pLKO.1-puro-DDR1 (TRCN0000121163), pLKO.1-puro-Sdc1 (TRCN0000302270), pLKO.1-puro-Nrf2 (TRCN0000054658) and pLKO.1-puro-Tfam (TRCN0000086064) were ordered from Sigma. pCDH-CMV-MCS-EF1-puro-Col1α1-6XHis and pLVX-IRES-Puro-NRF2$^{E79Q}$-Flag were made by Sangon Biotech (Shanghai, China). pLKO.1-blast-Ikkα, pLKO.1-puro-Nhe1, pLKO.1-puro-NHE1, pLKO.1-puro-NRF2, and lentiCRISPR v2-Puro-p62/*Sqstm1* have been described previously[11]. LentiCRISPR v2-Blast-ATG7 (ref. [29]) was a gift from S. Ghaemmaghami.

## Stable cell line construction

Lentiviral particles were generated as before[30]. MIA, 1305, KPC or KC cells and fibroblasts were transduced by combining 1 ml of viral particle-containing medium with 8 μg ml$^{-1}$ polybrene. The cells were fed 8 h later with fresh medium and selection was initiated 48 h after transduction using 1.25 μg ml$^{-1}$ puromycin or 10 μg ml$^{-1}$ blasticidin. IKKα$^{KD}$ KC, NRF2$^{KD}$ MIA and ATG7$^{Δ}$ MIA cells have been described previously[11].

## Mice

Female homozygous *Nu/Nu* nude mice and C57BL/6 mice were obtained at six weeks of age from Charles River Laboratories and The Jackson Laboratory, respectively. *Col1a1*$^{+/+}$ (Col I$^{WT}$) or *Col1a1*$^{r/r}$ (Col I$^{r/r}$) mice on a C57BL/6 background were obtained from D.B. at UCSD and were previously described[8,31]. Mice matched for age, gender and equal average tumour volumes were randomly allocated to different experimental groups on the basis of their genotypes. No sample size pre-estimation was performed but as many mice per group as possible were used to minimize type I/II errors. Both male and female mice were used unless otherwise stated. Blinding of mice was not performed except for IHC analysis. All mice were maintained in filter-topped cages on autoclaved food and water at constant temperature and humidity and in a pathogen-free controlled environment (23 °C ± 2 °C, 50–60%) with a standard 12-h light–12-h dark cycle. Experiments were performed in accordance with UCSD Institutional Animal Care and Use Committee and NIH guidelines and regulations. Animal protocol S00218 (M.K.) was approved by the UCSD Institutional Animal Care and Use Committee. The number of mice per experiment is indicated in the figure legends and their age is indicated in Methods.

## Orthotopic PDAC cell implantation

Col I$^{WT}$ or Col I$^{r/r}$ mice were pretreated with or without 50 μg kg$^{-1}$ CAE by intraperitoneal injections every hour, six times daily on the first, fourth and seventh days. On day 11, parental, NRF2$^{E79Q}$, DDR1$^{KD}$, DDR1$^{KD}$ + NRF2$^{E79Q}$, NRF2$^{KD}$ or TFAM$^{KD}$ KPC or KC cells were orthotopically injected into three-month-old Col I$^{WT}$ or Col I$^{r/r}$ mice as described[11]. After surgery, mice were given buprenorphine subcutaneously at a dose of 0.05–0.1 mg kg$^{-1}$ every 4–6 h for 12 h and then every 6–8 h for 3 additional days. Mice were analysed after four weeks.

## Intrasplenic PDAC cell implantation

Three-month-old Col I$^{WT}$ or Col I$^{r/r}$ mice were treated with or without an oral gavage of 25% CCl$_4$ in corn oil twice a week for two weeks. After two weeks of recovery, parental, NHE1$^{KD}$ or IKKα$^{KD}$ KPC or KC cells (10$^6$ cells in 50 μl phosphate-buffered saline; PBS) were adoptively transferred into the livers of Col I$^{WT}$ or Col I$^{r/r}$ mice by intrasplenic injection, followed by immediate splenectomy[10]. Mice were analysed 14 days after treatment with or without 10 mg kg$^{-1}$ EIPA (Sigma) by intraperitoneal injection every other day.

## Subcutaneous PDAC cell implantation

Homozygous BALB/c *Nu/Nu* female mice were injected subcutaneously in a single flank or in both flanks at 7 weeks of age with 5 × 10$^5$ parental, NHE1$^{KD}$, DDR1$^{KD}$ or DDR1$^{KD}$ + NRF2$^{E79Q}$ MIA cells or 1305 cells mixed with or without 5 × 10$^5$ wild-type, R/R, Col I$^{Δ}$ wild-type or Col I$^{Δ}$ R/R fibroblasts diluted 1:1 with BD Matrigel (BD Biosciences) in a total volume of 100 μl. Tumours were collected after four weeks. To evaluate the effect of IKKβ or mitochondrial protein synthesis inhibition on tumour growth, mice were treated with vehicle (dimethyl sulfoxide in PBS), ML120B (60 mg kg$^{-1}$) twice daily through oral gavage or tigecycline (50 mg kg$^{-1}$) twice daily through intraperitoneal injection for three weeks. Therapy was started one week after tumour implantation. Volumes (1/2 × (width$^2$ × length)) of subcutaneous tumours were calculated on the basis of digital caliper measurements. Mice were euthanized to avoid discomfort if the tumour diameter reached 2 cm.

## Samples of human PDAC

Survival analysis of patients expressing high and low levels of Col I–MMP was performed using The Cancer Genome Atlas (TCGA) data and the GEPIA2 platform. The collagen-cleaving signature consisted of MMP1, MMP2, MMP8, MMP9, MMP13 and MMP14. Overall survival was determined in the TCGA cohort of 178 patients with PDAC using a median cut-off.

A total of 106 specimens of human PDAC were acquired from patients who were diagnosed with PDAC between January 2017 and May 2021 at The Affiliated Drum Tower Hospital of Nanjing University Medical School. All patients received standard surgical resection and did not receive chemotherapy before surgery. Paraffin-embedded tissues were processed by a pathologist after surgical resection and confirmed as PDAC before further investigation. Overall survival duration was defined as the time from the date of diagnosis to that of death or last known follow-up examination. Survival information was available for 81 of the 106 patients. The study was approved by the Institutional Ethics Committee of The Affiliated Drum Tower Hospital with IRB 2021-608-01. Informed consent for tissue analysis was obtained before surgery. All research was performed in compliance with government policies and the Helsinki declaration.

## IHC

Pancreata or liver were dissected and fixed in 4% paraformaldehyde in PBS and embedded in paraffin. Five-micrometre sections were prepared and stained with H&E or sirius red. IHC was performed as before[11]. Slides were photographed on an upright light/fluorescent Imager A2 microscope with AxioVision Rel. 4.5 software (Zeiss). Antibody information is shown in Supplementary Table 2.

## IHC scoring

IHC scoring was performed as before[11]. Negative and weak staining was viewed as a low expression level and intermediate and strong staining was viewed as a high expression level. For cases with tumours with two satisfactory cores, the results were averaged; for cases with tumours with one poor-quality core, results were based on the interpretable core. On the basis of this evaluation system, a chi-squared test was used to estimate the association between the staining intensities of Col I–DDR1–NRF2 signalling proteins. The number of evaluated cases for each different staining in PDAC tissues and the scoring summary are indicated in Extended Data Fig. 7a.

## ECM preparation

Wild-type or R/R fibroblasts were seeded on 6, 12 or 96-well plates. One day after plating, cells were switched into DMEM (with pyruvate) with 10% dialysed FBS supplemented with or without 500 μM [$^3$H]-proline or [U-$^{13}$C]-glutamine and 100 μM vitamin C. Cells were cultured for five days with renewal of the medium every 24 h. Then fibroblasts were removed by washing in 1 ml or 500 μl or 100 μl per well PBS with 0.5% (v/v) Triton X-100 and 20 mM $NH_4OH$. The ECM was washed five times with PBS before cancer cell plating. The following day, cancer cells were switched into the indicated medium for 24 or 72 h.

## Cell imaging

Cells were cultured on coverslips coated with or without ECM and fixed in 4% paraformaldehyde for 10 min at room temperature or methanol for 10 min at −20 °C. Macropinosome visualization in cell and tissue and immunostaining were performed as previously described[11]. Images were captured and analysed using a TCS SPE Leica confocal microscope with Leica Application Suite AF 2.6.0.7266 software (Leica). Antibody information is shown in Supplementary Table 2.

## SHG

Mouse pancreatic tumour tissue was fixed in 4% paraformaldehyde in PBS and embedded in paraffin. Five-micrometre sections were prepared and deparaffinized in xylene, rehydrated in graded ethanol series as described[32], mounted using an aqueous mounting medium and sealed with a coverslip. All samples were imaged using a Leica TCS SP5 multiphoton confocal microscope and an HC APO LC 20× 1.00W was used throughout the experiment. The excitation wavelength was tuned to 840 nm, and a 420 ± 5-nm narrow bandpass emission controlled by a prism was used for detecting the SHG signal of collagen. SHG signal is generated when two photons of incident light interact with the non-centrosymmetric structure of collagen fibres, which leads to the resulting photons being half the wavelength of the incident photons. SHG measurements were performed using CT-Fire software (v.2.0 beta) (https://loci.wisc.edu/software/ctfire). The tumour area was confirmed by H&E staining.

## Immunoblotting and immunoprecipitation

Preparation of protein samples from cells and tissues, immunoblotting and immunoprecipitation were performed as before[10,30]. Immunoreactive bands were detected by an automatic X-ray film processor or a KwikQuant Imager. Antibody information is shown in Supplementary Table 2.

## Chromatin immunoprecipitation

Cells were cross-linked with 1% formaldehyde for 10 min and the reaction was stopped with 0.125 M glycine for 5 min. The chromatin immunoprecipitation assay was performed as described[11]. Cells were lysed and sonicated on ice to generate DNA fragments with an average length of 200–800 bp. After pre-clearing, 1% of each sample was saved as the input fraction. Immunoprecipitation was performed using antibodies that specifically recognize NRF2 (CST, 12721). DNA was eluted and purified from complexes, followed by PCR amplification of the target promoters or genomic loci using primers for mouse *Tfam*: 5′-GAGGCAGGGTCTCATG-3′ and 5′-CAAGCTGAGTTCTATC-3′; 5′- TCTGGGCCATCTTGGG-3′ and 5′- CCATGGGCCTGGGCTG-3′.

## Quantitative PCR analysis

Total RNA and DNA were extracted using the All Prep DNA/RNA Mini Kit (Qiagen). RNA was reverse-transcribed using a Superscript VILO cDNA synthesis kit (Invitrogen). Quantitative (q)PCR was performed as described[11]. Primers obtained from the NIH Primer-BLAST (https://www.ncbi.nlm.nih.gov/tools/primer-blast/index.cgi?LINK_LOC=BlastHome) are shown in Supplementary Table 3.

## RNA-seq library preparation, processing and analysis

Total RNA was isolated as described above from KPC samples grown on wild-type ($n = 3$) or R/R ($n = 3$) ECM as indicated. RNA purity was assessed by an Agilent 2100 Bioanalyzer. Five hundred nanograms of total RNA was enriched for poly-A-tailed RNA transcripts by double incubation with Oligo d(T) Magnetic Beads (NEB, S1419S) and fragmented for 9 min at 94 °C in 2× Superscript III first-strand buffer containing 10 mM DTT (Invitrogen, P2325). The reverse-transcription reaction was performed at 25 °C for 10 min followed by 50 °C for 50 min. The reverse-transcription product was purified with RNAClean XP (Beckman Coulter, A63987). Libraries were ligated with dual unique dual index (UDI) (IDT) or single UDI (Bioo Scientific), PCR-amplified for 11–13 cycles, size-selected using one-sided 0.8× AMPure clean-up beads, quantified using the Qubit dsDNA HS Assay Kit (Thermo Fisher Scientific) and sequenced on a HiSeq 4000 or NextSeq 500 (Illumina).

RNA-seq reads were aligned to the mouse genome (GRCm38/mm10) using STAR. Biological and technical replicates were used in all experiments. Quantification of transcripts was performed using HOMER (v.4.11). Principal component analysis (PCA) was obtained on the basis of transcripts per kilobase million (TPM) on all genes from all samples. Expression value for each transcript was calculated using the analyzeRepeats.pl tool of HOMER. Differential expression analysis was calculated using getDiffExpression.pl tool of HOMER. Pathway analyses were performed using the Molecular Signature Database of GSEA.

## scRNA-seq analysis

Samples from five primary tumours from patients with PDAC and one PDAC liver metastasis were obtained[33] and analysed separately to better identify cell heterogeneity and clusters. The datasets were processed in R (v.4.0.2) and Seurat[34] (v.4.0.5) and cells with at least 200 genes and genes expressed in at least 3 cells were retained for further quality control analysis for the percentage of mitochondrial genes expressed, total genes expressed and unique molecular identifier (UMI) counts. The gene–cell barcode matrix obtained after quality control analysis was log-normalized and 3,000 variable genes were identified and scaled to perform PCA. The five PDAC primary patient samples were then batch-corrected and integrated using a reciprocal PCA (RPCA) pipeline in Seurat using 'FindIntegrationAnchors' and 'IntegrateData' functions. The 'integrated' assay was again scaled to perform PCA. The top significant principal components of PCA were identified using 'ElbowPlot' in each dataset. To cluster and visualize the cells, 'FindNeighbours', 'FindClusters' and 'RunUMAP' functions were used on the top identified principal components in each dataset.

The cell types were identified by manual annotation of well-known makers[33], namely: epithelial-tumour cells (*EPCAM* and *KRT8*), pancreatic epithelial cells (*CPA1* and *CTRB1*), T cells (*CD3D* and *IL7R*), myeloid cells (*CD14*, *CD68*, *FCGR3A* and *LYZ*), NK cells (*NKG7* and *GNLY*), B cells (*CD79A* and *MS4A1*), dendritic cells (*FCGR1A* and *CPA3*), endothelial cells (*PECAM1*, *KDR* and *CDH5*), fibroblasts (*ACTA2*, *COL1A1*, *COLEC11* and *DCN*), vascular smooth muscle cells (*MYH11* and *ACTA2*), hepatocytes (*ALB*, *APOE* and *CPS1*), cholangiocytes (*ANXA4*, *KRT7* and *SOX9*), plasma cells (*JCHAIN* and *IGKC*) and cycling cells (*TOP2A* and *MKI67*).

M1/M2 macrophages were designated as described[35]: M1-like macrophages (*AZIN1*, *CD38*, *CXCL10*, *CXCL9*, *FPR2*, *IL18*, *IL1B*, *IRF5*, *NIFKBIZ*, *TLR4*, *TNF* and *CD80*) and M2-like macrophages (*ALOX5*, *ARG1*, *CHIL3*, *CD163*, *IL10*, *IL10RA*, *IL10RB*, *IRF4*, *KIF4*, *MRC1*, *MYC*, *SOCS2* and *TGM2*). The mean expression score for the M1 and M2 signatures were computed for each macrophage subcluster using 'AddModuleScore' function and clusters with a higher M1 or M2 signature score were assigned M1-like or M2-like annotation, respectively.

### Metabolite extraction and analysis
Cells grown on a 12-well plate coated with or without ECM. Metabolite extraction and analysis were performed as before[11]. Gas chromatography-mass spectrometry (GC-MS) analysis was performed using an Agilent 6890 gas chromatograph equipped with a 30-m DB-35MS capillary column connected to an Agilent 5975B mass spectrometer operating under electron impact ionization at 70 eV. For measurement of amino acids, the gas chromatograph oven temperature was held at 100 °C for 3 min and increased to 300 °C at 3.5 °C per min. The mass spectrometer source and quadrupole were held at 23 °C and 150 °C, respectively, and the detector was run in scanning mode, recording ion abundance in the range of 100–605 *m/z*. Mole per cent enrichments of stable isotopes in metabolite pools were determined by integrating the appropriate ion fragments and correcting for natural isotope abundance as previously described[36].

### Cell viability assay
Cells were plated in 96-well plates coated with or without ECM at a density of 3,000 cells (MIA, 1305) or 1,500 cells (KPC or KC) per well and incubated overnight before treatment. 7rh (500 nM), ML120B (10 μM), EIPA (10.5 μM), IPI549 (600 nM), MBQ-167 (500 nM), MRT68921 (600 nM) or ML385 (10 μM), or their combinations, were added to the wells in the presence of complete medium (CM), LG medium or LQ medium for 72 h. Cell viability was determined with a Cell Counting Kit-8 assay (Glpbio). Optical density was read at 450 nm and analysed using a microplate reader with SoftMax 6.5 software (FilterMax F5, Molecular Devices). For all experiments, the medium was replaced every 24 h.

### Luminescence ATP detection assay
KPC or KC cells were grown on 96-well plates coated with or without the indicated ECM in the presence of 100 μl CM or LG medium with or without EIPA (10.5 μM), MBQ-167 (500 nM), MRT68921 (600 nM) or their combinations for 24 h. Then the cell number was measured. Intracellular ATP was determined with a luminescence ATP detection assay system (PerkinElmer) according to the manufacturer's protocol. Finally, luminescence was measured and normalized to cell number.

### L-amino acid assay
KPC or KC cells were grown on six-well plates coated with or without the indicated ECM in the presence of 100 μl LG medium with or without EIPA (10.5 μM), MRT68921 (600 nM) or their combinations for 24 h. Total amounts of free L-amino acids (except for glycine) were measured using an L-Amino Acid Assay Kit (Colorimetric, antibodies) according to the manufacturer's protocol. The concentration of L-amino acids was calculated within samples by comparing the sample optical density to the standard curve and normalized to cell number.

### Statistics and reproducibility
Macropinosomes or mitochondria were quantified by using the 'Analyze Particles' feature in Image J (NIH). Macropinocytotic uptake index[37] or mitochondria number was computed by the macropinosome or mitochondria area in relation to the total cell area for each field and then by determining the average across all the fields (six fields). Tumour area (%) was quantified by using the 'Polygon' and 'Measure' feature in Fiji Image J and was computed by tumour area in relation to total area for each field and then by determining the average across all the fields (five fields). Positive area of protein expression in tumour (%) was quantified by using 'Colour Deconvolution', 'H DAB', and 'Analyze Particles' feature in Fiji Image J and was computed by the protein-positive area in relation to the tumour area for each field and then by determining the average across all the fields (5–6 fields). These measurements were done on randomly selected fields of view. A two-tailed unpaired Student's *t*-test was performed for statistical analysis using GraphPad Prism software. Data are presented as mean ± s.e.m. Kaplan–Meier survival curves were analysed by log-rank test. Statistical correlation between Col I–DDR1–NRF2 signalling proteins in human PDAC specimens was determined by two-tailed chi-squared test. (****$P < 0.0001$, ***$P < 0.001$, **$P < 0.01$ and *$P < 0.05$). All experiments except the IHC analysis of 106 human specimens were repeated at least 3 times.

### Reporting summary
Further information on research design is available in the Nature Research Reporting Summary linked to this article.

### Data availability
RNA-seq data are available at the Gene Expression Omnibus (GEO) under accession number GSE206218. scRNA-seq data were obtained from a published GEO dataset (GSE156405). Graph data and raw images of immunoblot and DNA gels are provided within the Source Data. All raw image data including immunostaining, immunoblotting, DNA gels, IHC, H&E and sirius red staining were uploaded to Mendeley Data (https://doi.org/10.17632/9v2hyb4j7n.1). Source data are provided with this paper.

### Code availability
Custom computer code used in the scRNA-seq analysis is available at https://github.com/ajynair/Collagen_DDR1_PDACmets.

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

**Acknowledgements** We thank members of the M.K. laboratory for discussions; Cell Signaling Technologies, Santa Cruz Technologies, Thermo Fisher Scientific, Promega and

MedChemExpress for gifts of antibodies and reagents; and UCSD Tissue Technology Shared Resources (TTSR) and J. Santini at the UCSD School of Medicine Microscopy Core for histology and microscopy services. Research was supported by grants from Padres Pedal the Cause/C3 (PPTC2018 to M.K. and A.M.L.); the Youth Program of the National Natural Science Foundation of China (81802757 to H.S. and 82002931 to F.Y.); the NIH (R01CA211794, R37AI043477, P01DK098108 and U01AA027681 to M.K., U01CA274295 to M.K., R.F.S. and A.M.L., R01CA155630 to A.M.L. and R01CA234245 and R01CA218254 to C.M.M.); the National Key Research and Development Program of China (2016YFC0905900 to B.S.); the National Cancer Institute Cancer Center Support Grant (CCSG) (P30CA23100 to TTSR); and the UCSD School of Medicine Microscopy Core (NINDS P30-NS047101). Additional support was provided by Ride the Point (M.K. and A.M.L.); the Research for a Cure of Pancreatic Cancer Fund and the Alexandrina M. McAfee Trust Foundation (A.M.L.); and the UC Pancreatic Cancer Consortium to A.M.L., who is the Homer T. Hirst III Professor of Oncology in Pathology, and M.K., who is the Ben and Wanda Hillyard Chair for Mitochondrial and Metabolic Diseases.

**Author contributions** M.K. and H.S. conceived the project. H.S. designed the study and H.S. and F.Y. performed most experiments. F.Y., H.S. and R.F. performed IHC analysis of human and mouse PDAC. H.S., F.Y., B.T., J. Siruno., M.L., Y.L. and N. Sinchai performed immunoblotting and qPCR analysis. C.M.M. and A.K. performed the $^{13}$C-tracing experiments. A.M., N. Sun. and S.D. performed Seahorse experiments. J.L. performed several orthotopic PDAC cell implantations. J.B. performed several intrasplenic PDAC implantations. R.F.S., A.N. and A.F. performed scRNA-seq analysis. S.B.R. performed RNA-seq analysis. J. Santini detected SHG signal of collagen. D.A.B. provided Col I$^{r/r}$ mice. A.M.L. provided human PDAC 1305 cells. B.S. collected human PDAC tissue and supervised and supported F.Y. and R.F. M.K. and H.S. wrote the manuscript, with all authors contributing and providing feedback and advice.

**Competing interests** The authors declare no competing interests.

## Additional information
**Correspondence and requests for materials** should be addressed to Beicheng Sun or Michael Karin.

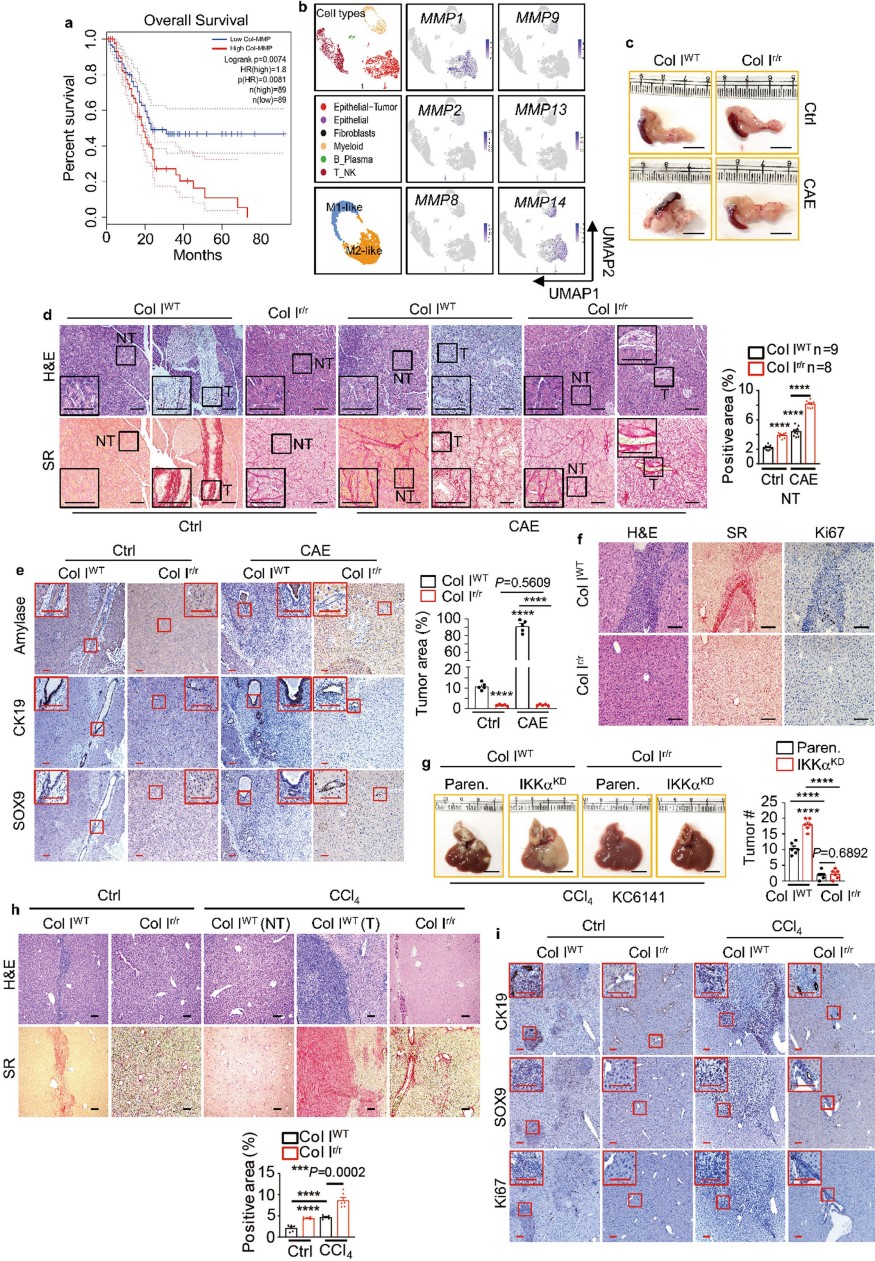

**Extended Data Fig. 1 | Col I cleavage stimulates PDAC growth. a**, Overall survival of patients with PDAC from TCGA with high and low collagen-cleaving MMP signature (MMP1, 2, 8, 9, 13, 14). Significance was analysed by log-rank test. **b**, UMAPs showing scRNA-seq data from 5 primary PDACs, displaying cell types and expression of the most abundant *MMP* mRNAs. **c**, Pancreas morphology 4 wk after orthotopic KPC cell transplantation into Col I^WT or Col I^r/r mice −/+ CAE pretreatment. **d**, H&E and sirius red (SR) staining of pancreatic sections from above mice. Boxed areas were further magnified. Quantification of SR positivity in nontumor (NT) areas is shown to the right. **e**, IHC of pancreatic sections from above mice. Quantification of tumour areas

is shown to the right. **f**, H&E, SR, Ki67 staining of liver sections from above CAE-pretreated mice. **g**, Liver gross morphology and tumour numbers (#) 2 wk after i.s. transplantation of Paren. or IKKα knockdown (KD) KC cells into CCl₄ pretreated Col I^WT or Col I^r/r mice. **h**, H&E and SR staining of liver sections 2 wk after i.s. transplantation of KPC cells into Col I^WT and Col I^r/r mice −/+ CCl₄ pretreatment. Quantification of SR positivity in NT areas is shown at the bottom. **i**, IHC of liver sections from above mice. Boxed areas show higher magnification. Results in (**e**) (*n* = 5 fields), (**g**), (**h**) (*n* = 6 mice) and (**d**) are mean ± s.e.m. Statistical significance determined by two-tailed *t*-test. \*\*\**P* < 0.001, \*\*\*\**P* < 0.0001. Scale bars in (**d**–**f**, and **h**,**i**), 100 μm, (**c**,**g**), 1 cm.

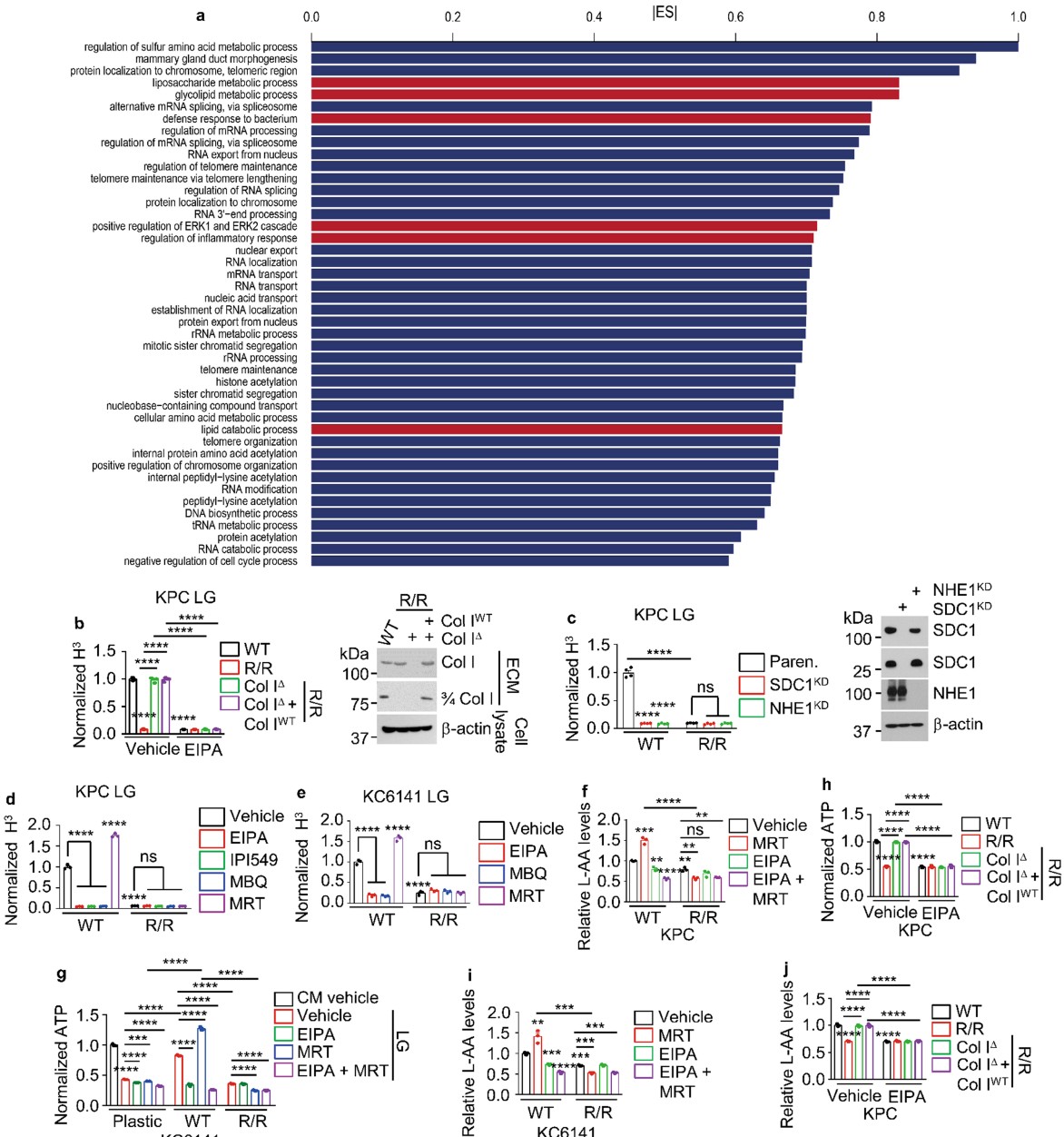

**Extended Data Fig. 2 | The Col I state controls PDAC gene expression and metabolism. a**, Dataset enrichment of RNA-seq data (*n* = 3) from KPC cells plated on wild-type (WT) (blue) or R/R (red) ECM and incubated in LG for 24 h. **b**, KPC cells grown on [³H]-proline-labelled WT, R/R, Col Iᐃ R/R or Col Iᐃ R/R + Col Iᵂᵀ ECM were incubated in LG −/+ EIPA for 24 h. [³H] uptake is presented relative to vehicle treated WT ECM-plated KPC cells. IB analysis of iCol I and 3/4 Col I in ECM produced by indicated fibroblasts. **c**, Indicated KPC cells were plated on [³H]-proline-labelled ECM and incubated in LG for 24 h. [³H] uptake is presented relative to Paren. uptake. KD efficiency is demonstrated. **d**, KPC cells were plated on [³H]-proline-labelled ECM and incubated in LG −/+ indicated reagents for 24 h. [³H] uptake is presented as above. **e**, [³H] uptake by KC cells treated as above. **f**, AA content of ECM-plated KPC cells incubated in LG −/+ indicated

reagents for 24 h. Cell number normalized data are presented relative to untreated WT ECM-plated cells. **g**, KC cells were plated −/+ WT or R/R ECM and incubated in complete (CM) or LG media −/+ indicated reagents for 24 h. Cellular ATP content is presented relative to untreated plastic-plated cells. **h**, KPC cells were plated as in (**b**) and incubated in LG −/+ EIPA for 24 h. Cellular ATP content is presented relative to untreated WT ECM-plated cells. **i**, Total AA in KC cells plated on ECM and incubated in LG −/+ indicated reagents for 24 h. Data are presented as above. **j**, Total AA in KPC cells treated as in (**h**). Results in (**b,d–j**) (*n* = 3 independent experiments), (**c**) (*n* = 4 independent experiments) are mean ± s.e.m. Statistical significance determined by two-tailed *t*-test. **P < 0.01, ***P < 0.001, ****P < 0.0001. Exact P values in (**b–j**) are shown in Source Data.

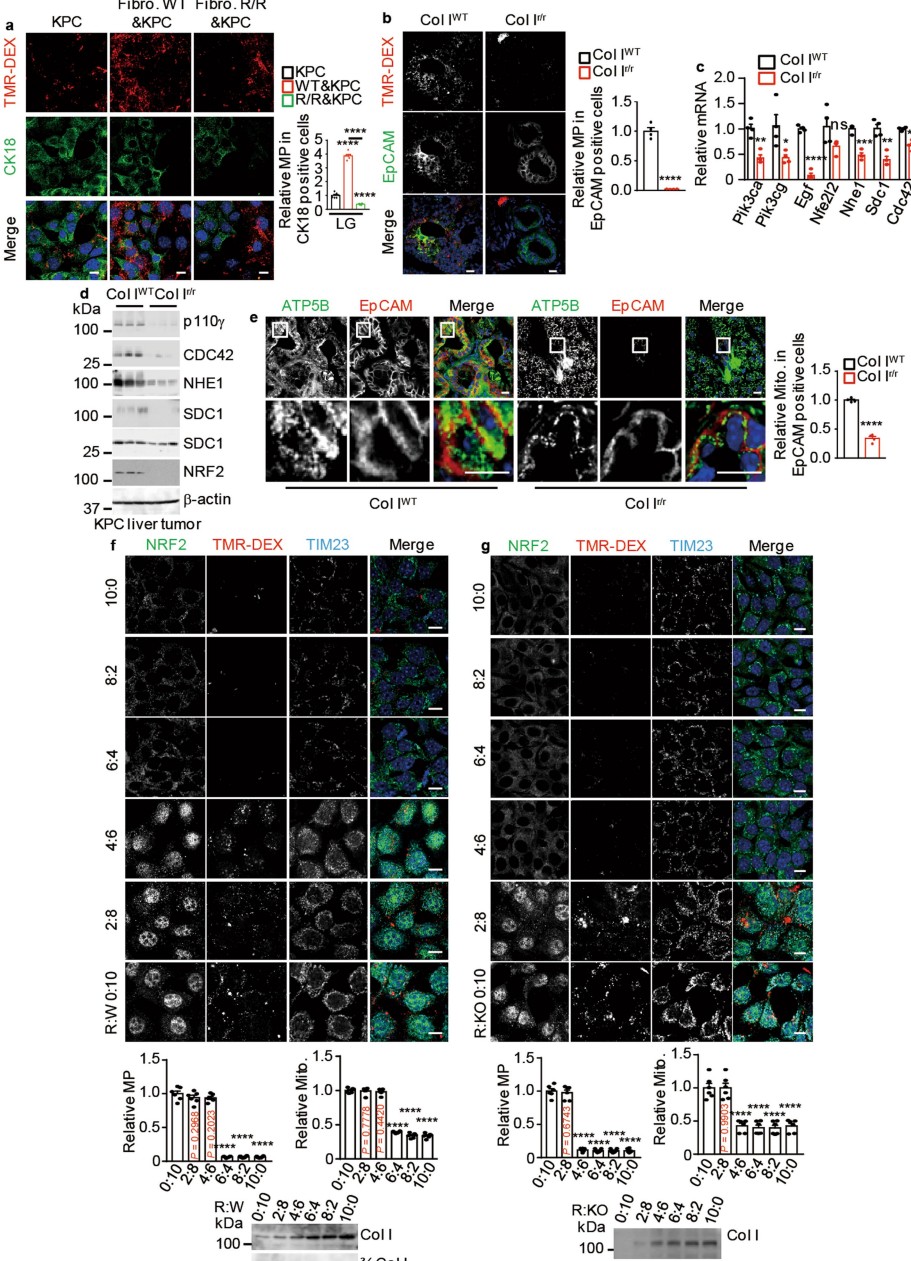

**Extended Data Fig. 3 | The cleaved to intact Col I ratio controls macropinocytosis and mitochondrial biogenesis. a**, Macropinocytosis (MP) visualization and quantification using TMR-DEX in KPC cells co-cultured with WT or R/R fibroblasts and incubated in LG medium for 24 h. KPC cells were marked by cytokeratin 18 (CK18, green). **b**, Representative images, and MP quantification in TMR-DEX-injected pancreatic tissue from Col I$^{WT}$ or Col I$^{r/r}$ mice 4 wk after orthotopic KPC cell transplantation. Carcinoma cells are marked by EpCAM staining (green). Quantification is on the right. **c**, qRT-PCR analysis of MP-related mRNAs in liver tumours 2 wk after i.s. KPC cell transplantation into CCl$_4$ pretreated Col$^{WT}$ or Col I$^{r/r}$ mice. Exact $P$ values are shown in Source Data. **d**, IB analysis of MP-related proteins in above liver

tumours. **e**, Representative images, and quantification of Mito. (ATP5B, green) in pancreatic tissue from indicated mice analysed as in (**b**). Carcinoma cells are marked by EpCAM staining (red). Quantification is on the right. **f,g**, Representative images, and quantification of Mito. (TIM23) and MP in TMR-DEX-incubated KPC cells grown on mixed ECM produced by R/R and WT (R:W) (**f**) or R/R and Col I$^{Δ}$R/R (R:KO) (**g**) fibroblasts in the indicated ratios and incubated in LG medium for 24 h. IB analysis of iCol I or cCol I (3/4 Col I) in above ECM preparations is shown at the bottom. Results in (**a,f,g**) ($n$ = 6 fields), (**b,c,e**) ($n$ = 4 mice) are mean ± s.e.m. Statistical significance determined by two-tailed $t$-test. *$P$ < 0.05, **$P$ < 0.01, ***$P$ < 0.001, ****$P$ < 0.0001. Scale bars, 10 μm.

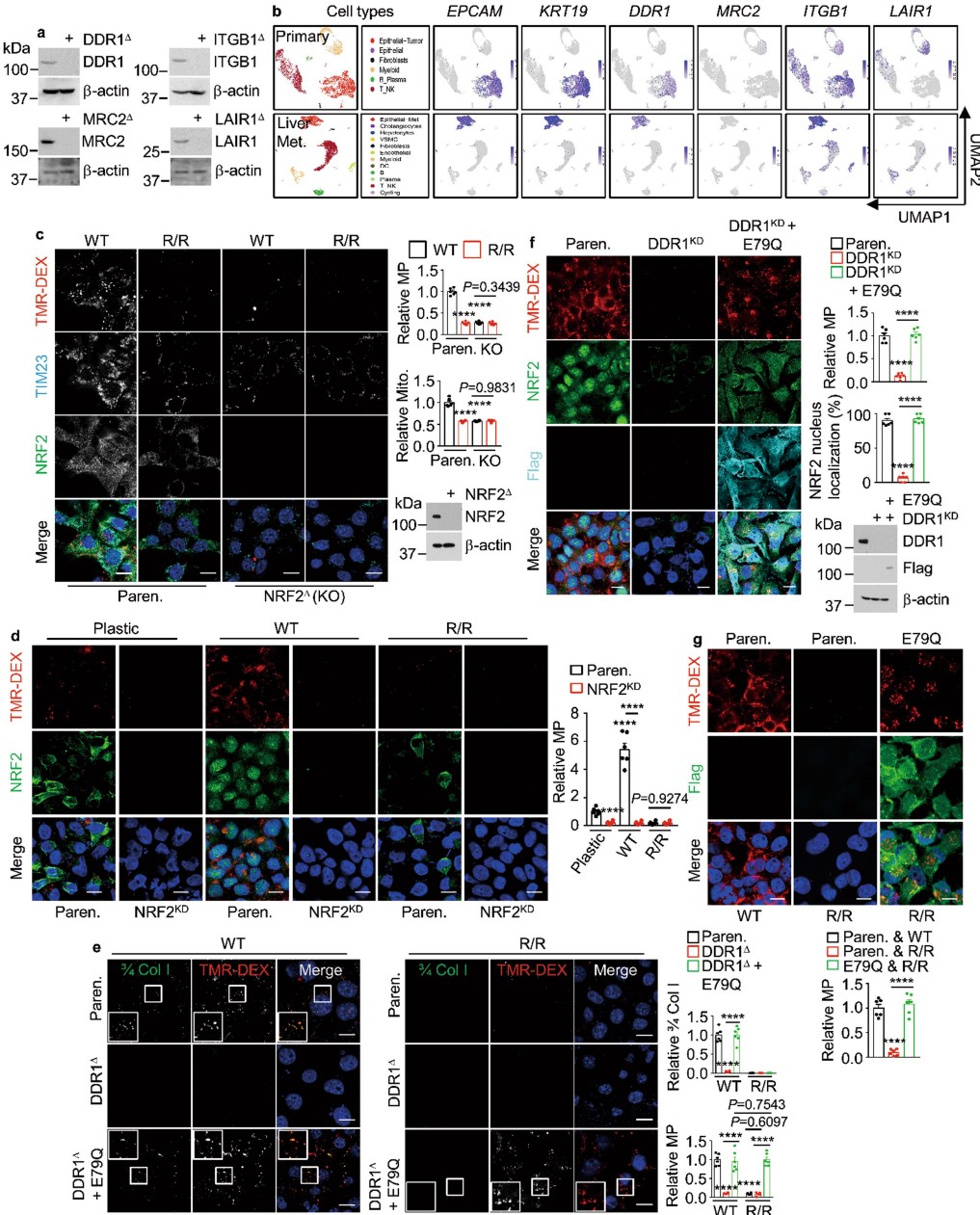

**Extended Data Fig. 4 | Col I controls macropinocytosis and mitochondrial content through DDR1–NRF2 signalling. a**, IB analysis of KPC cells ablated for indicated collagen receptors. **b**, UMAPs showing scRNA-seq data from 5 primary PDAC (upper row) and 1 PDAC liver metastasis (lower row), displaying the identified cell populations and expression of the indicated mRNAs. **c**, Representative images, and quantification of Mito. and MP in TMR-DEX-incubated Paren. and NRF2$^\Delta$ (KO) KPC cells plated on WT or R/R ECM and incubated in LG for 24 h. NRF2 IB analysis is shown on the right. **d**, MP and NRF2 localization in Paren. and NRF2$^{KD}$ MIA cells plated –/+ WT or R/R ECM and

incubated in LG for 24 h. MP quantification is shown on the right. **e**, 3/4 Col I and MP imaging and quantification in KPC cells treated as above. Although NRF2$^{E79Q}$ (E79Q) stimulates MP, cCol I uptake is detected only in cells plated on WT ECM. **f**, Representative images, and quantification of MP and nuclear NRF2 in 1305 cells plated on WT ECM and incubated in LG for 24 h. IB analysis of DDR1 and Flag-tagged E79Q is shown on the right. **g**, MP imaging and quantification in Paren. and E79Q MIA cells plated on WT or R/R ECM and incubated in LG for 24 h. Results in (**c-g**) (*n* = 6 fields) are mean ± s.e.m. Statistical significance was determined by two-tailed *t*-test. ****$P$ < 0.0001. Scale bars, 10 μm.

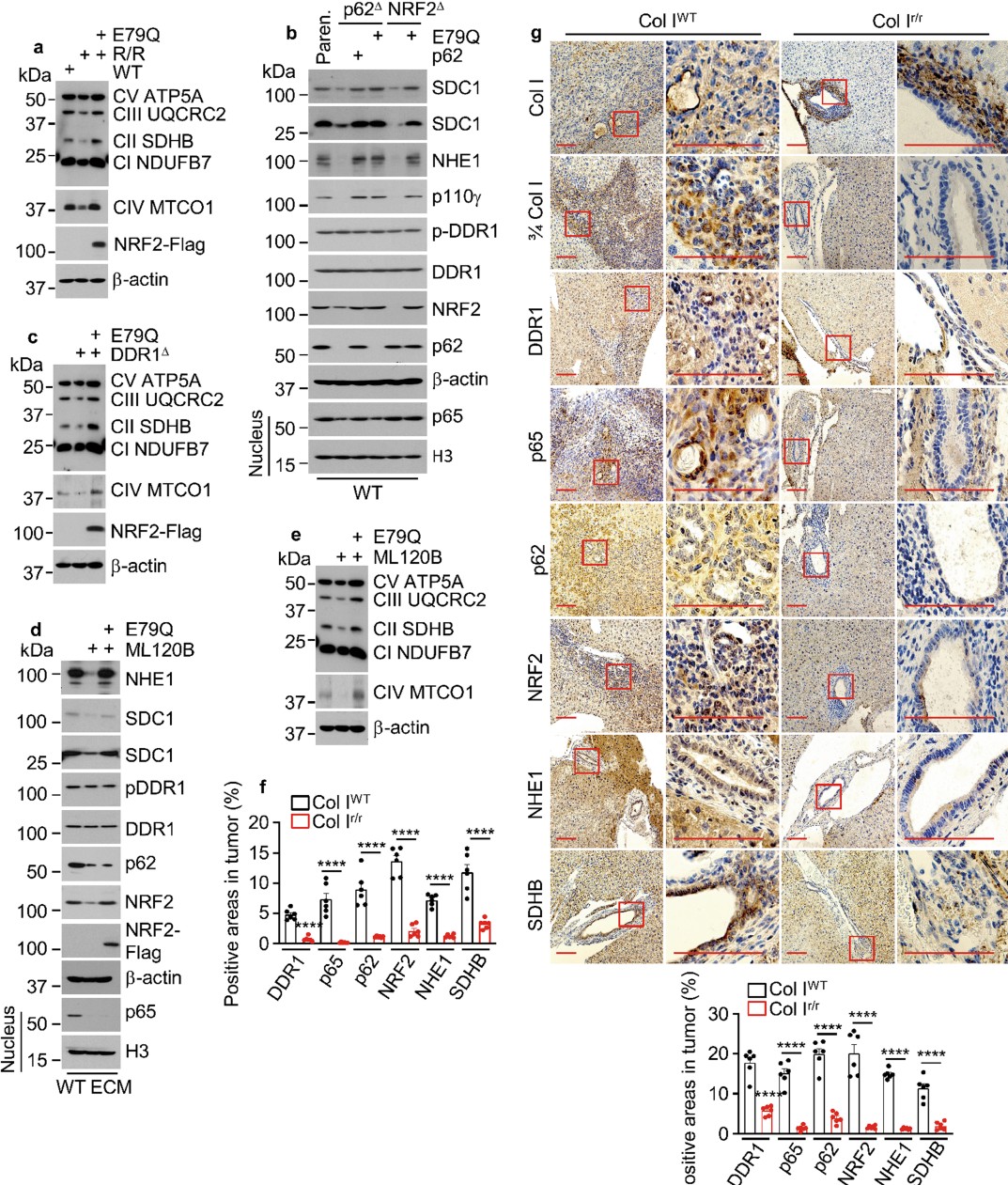

**Extended Data Fig. 5 | cCol I–DDR1–NRF2 signalling controls macropinocytosis and mitochondrial protein expression. a**, IB analysis of ETC complexes I-V (CI-CV) in Paren. and E79Q KPC cells plated on WT or R/R ECM and incubated in LG for 24 h. **b**, IB analysis of indicated KPC cells –/+ ectopic p62 or E79Q plated on WT ECM and incubated in LG for 24 h. **c**, IB of ETC proteins in Paren., DDR1$^\Delta$, or E79Q/DDR1$^\Delta$ KPC cells plated on WT ECM and incubated in LG for 24 h. **d,e**, IB of indicated proteins in Paren. or E79Q KPC cells plated on WT ECM and incubated in LG medium –/+ML120B for 24 h. **f**, Staining intensity of the indicated proteins in tumour areas depicted Fig. 4e determined with Image J. **g**, IHC of liver sections prepared 2 wk after i.s. transplantation of KPC cells into $CCl_4$ pretreated Col I$^{WT}$ and Col I$^{r/r}$ mice. Scale bars, 100 μm. Image J determined staining intensity of indicated proteins in tumour areas is shown at the bottom. Results in (**f,g**) (*n* = 6 fields) are mean ± s.e.m. Statistical significance determined by two-tailed *t*-test. ****$P < 0.0001$.

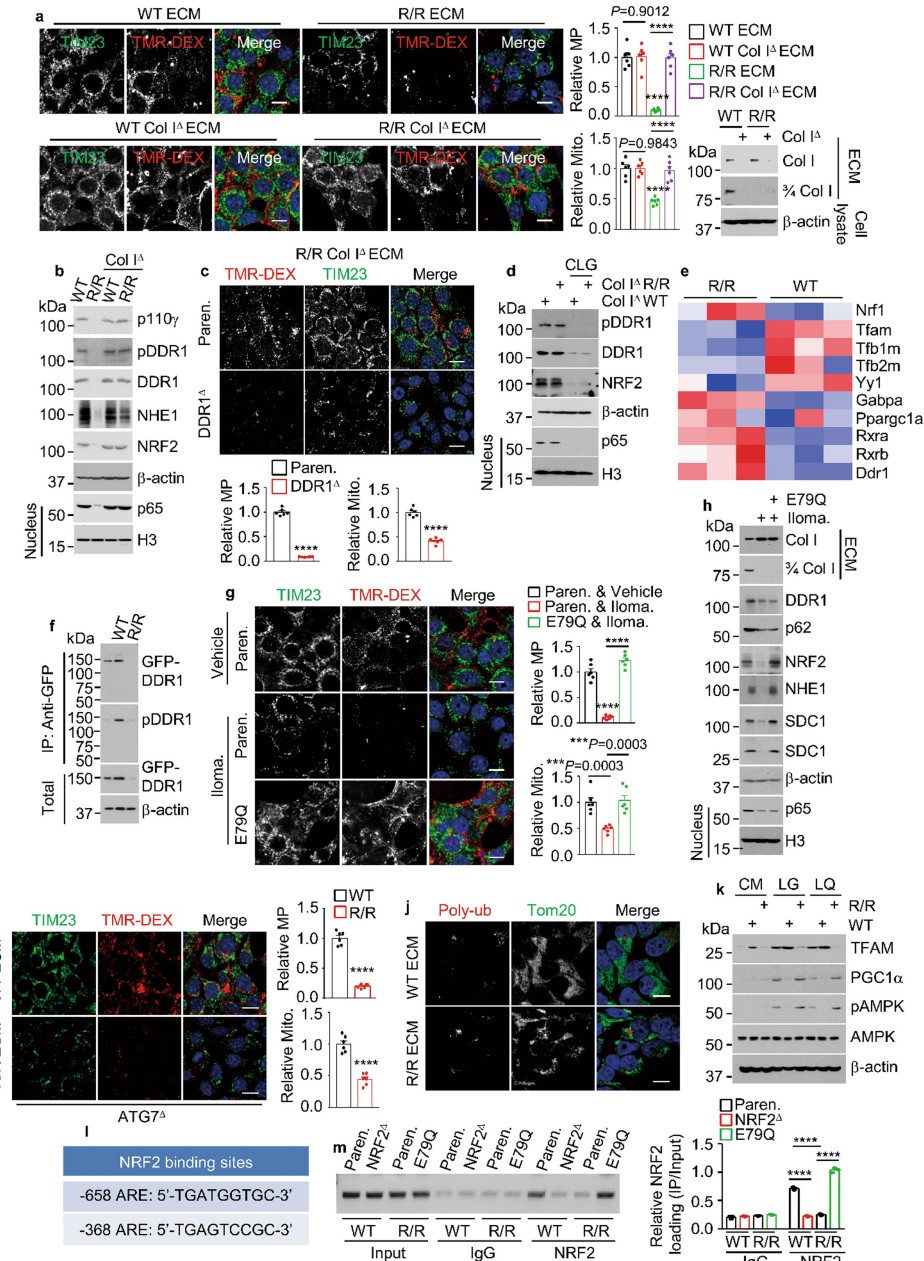

**Extended Data Fig. 6 | Inhibition of Col I cleavage shuts down NRF2-driven macropinocytosis and mitochondrial biogenesis. a**, MP and mitochondria in KPC cells plated on indicated ECM and incubated in LG for 24 h. Col I IB in indicated ECM preparations is shown on the right. **b**, IB of indicated proteins in above cells. **c**, MP and mitochondria in indicated KPC cells plated on R/R Col I$^\Delta$ ECM and incubated in LG for 24 h. **d**, IB of indicated proteins in 1305 cells plated on ECM produced by bacterial collagenase (CLG, 50 µg/ml) treated or untreated fibroblasts and incubated in LG medium −/+ CLG for 24 h. **e**, Genes differentially expressed between KPC cells plated on indicated ECM. Blue: replicates with low expression (z-score = −2); red: replicates with high expression (z-score = +2). **f**, Immunoprecipitation (IP) of GFP–DDR1 from 1305 cells plated −/+ indicated ECM. **g**, MP and mitochondria in Paren. or E79Q KPC cells plated on ECM produced by Ilomastat (Iloma.) treated or untreated WT

fibroblasts and incubated in LG medium −/+ Iloma. for 24 h. **h**, IB of indicated proteins in above cells. **i**, MP and mitochondria in ATG7$^\Delta$ MIA PaCa-2 cells plated on indicated ECM and incubated in LG for 24 h. **j**, Imaging of 1305 cells plated on indicated ECM showing rare poly-Ub and Mito. (Tom20) colocalization. **k**, IB analysis of KPC cells plated −/+ indicated ECM and incubated in indicated media for 24 h. **l**, Locations of putative NRF2-binding sites (AREs) relative to the transcriptional start site (TSS, +1) of the mouse *Tfam* gene. **m**, Chromatin IP probing NRF2 recruitment to the *Tfam* promoter in KPC cells plated on WT or R/R ECM and incubated in LG for 24 h. The image shows PCR-amplified ARE-containing promoter DNA fragments. Quantitation on the right. Results in (**a,c,g,i**) (*n* = 6 fields), (**m**) (*n* = 3 independent experiments) are mean ± s.e.m. Statistical significance determined by two-tailed *t*-test. \*\*\**P* < 0.001, \*\*\*\**P* < 0.0001. Scale bars, 10 µm.

**a**

| | Low | High |
|---|---|---|
| Masson | 26 | 80 |
| Col I | 13 | 93 |
| 3/4 Col I | 29 | 77 |
| DDR1 | 39 | 67 |
| p65 | 42 | 64 |
| NRF2 | 36 | 70 |
| NHE1 | 47 | 59 |
| SDC1 | 45 | 61 |
| CDC42 | 42 | 64 |
| SDHB | 30 | 76 |
| a-SMA | 29 | 77 |
| MMP1 | 46 | 60 |

**b**

$P < 0.0001$ ****

| | DDR1 high | DDR1 low | Total |
|---|---|---|---|
| cCol I high | 58 | 19 | 77 |
| cCol I low | 9 | 20 | 29 |
| Total | 67 | 39 | 106 |

$P = 0.0002$ ***

| | p65 high | p65 low | Total |
|---|---|---|---|
| cCol I high | 55 | 22 | 77 |
| cCol I low | 9 | 20 | 29 |
| Total | 64 | 42 | 106 |

$P < 0.0001$ ****

| | NRF2 high | NRF2 low | Total |
|---|---|---|---|
| cCol I high | 60 | 17 | 77 |
| cCol I low | 10 | 19 | 29 |
| Total | 70 | 36 | 106 |

$P = 0.0001$ ***

| | SDC1 high | SDC1 low | Total |
|---|---|---|---|
| cCol I high | 53 | 24 | 77 |
| cCol I low | 8 | 21 | 29 |
| Total | 61 | 45 | 106 |

$P = 0.0141$ *

| | CDC42 high | CDC42 low | Total |
|---|---|---|---|
| cCol I high | 52 | 25 | 77 |
| cCol I low | 12 | 17 | 29 |
| Total | 64 | 42 | 106 |

$P = 0.001$ **

| | SDHB high | SDHB low | Total |
|---|---|---|---|
| cCol I high | 62 | 15 | 77 |
| cCol I low | 14 | 15 | 29 |
| Total | 76 | 30 | 106 |

$P = 0.0002$ ***

| | MMP1 high | MMP1 low | Total |
|---|---|---|---|
| cCol I high | 52 | 25 | 77 |
| cCol I low | 8 | 21 | 29 |
| Total | 60 | 46 | 106 |

$P = 0.9743$

| | α-SMA high | α-SMA low | Total |
|---|---|---|---|
| cCol I high | 56 | 21 | 77 |
| cCol I low | 21 | 8 | 29 |
| Total | 77 | 29 | 106 |

$P = 0.007$ **

| | p65 high | p65 low | Total |
|---|---|---|---|
| DDR1 high | 47 | 20 | 67 |
| DDR1 low | 17 | 22 | 39 |
| Total | 64 | 42 | 106 |

$P < 0.0001$ ****

| | NRF2 high | NRF2 low | Total |
|---|---|---|---|
| DDR1 high | 54 | 13 | 67 |
| DDR1 low | 16 | 23 | 39 |
| Total | 70 | 36 | 106 |

$P = 0.0002$ ***

| | NRF2 high | NRF2 low | Total |
|---|---|---|---|
| p65 high | 51 | 13 | 64 |
| p65 low | 19 | 23 | 42 |
| Total | 70 | 36 | 106 |

$P = 0.0127$ *

| | NHE1 high | NHE1 low | Total |
|---|---|---|---|
| NRF2 high | 45 | 25 | 70 |
| NRF2 low | 14 | 22 | 36 |
| Total | 59 | 47 | 106 |

$P = 0.0014$ **

| | SDC1 high | SDC1 low | Total |
|---|---|---|---|
| NRF2 high | 48 | 22 | 70 |
| NRF2 low | 13 | 23 | 36 |
| Total | 61 | 45 | 106 |

$P = 0.0471$ *

| | CDC42 high | CDC42 low | Total |
|---|---|---|---|
| NRF2 high | 47 | 23 | 70 |
| NRF2 low | 17 | 19 | 36 |
| Total | 64 | 42 | 106 |

$P = 0.0285$ *

| | SDHB high | SDHB low | Total |
|---|---|---|---|
| NRF2 high | 55 | 15 | 70 |
| NRF2 low | 21 | 15 | 36 |
| Total | 76 | 30 | 106 |

**Extended Data Fig. 7 | Correlation between Col I–DDR1–NRF2 signalling components and inflammation in human PDAC. a**, Numbers and percentages of human PDAC specimens ($n = 106$ specimens) positive for the indicated proteins (arbitrarily indicated as low and high). **b**, Correlation between indicated proteins in above specimens was analysed by a two-tailed Chi-square test. *$P < 0.05$, **$P < 0.01$, ***$P < 0.001$, ****$P < 0.0001$. cCol I (3/4 Col I). **c**, Representative IHC and quantification of the indicated markers in cCol I$^{high}$ (#43) and cCol I$^{low}$ (#54) human PDAC specimens from Fig. 5a. Mean ± s.e.m. ($n = 16$ specimens). Statistical significance determined by two-tailed $t$-test. ***$P = 0.0001$. *$P = 0.0145$, *$P = 0.0388$. Scale bars, 100 μm.

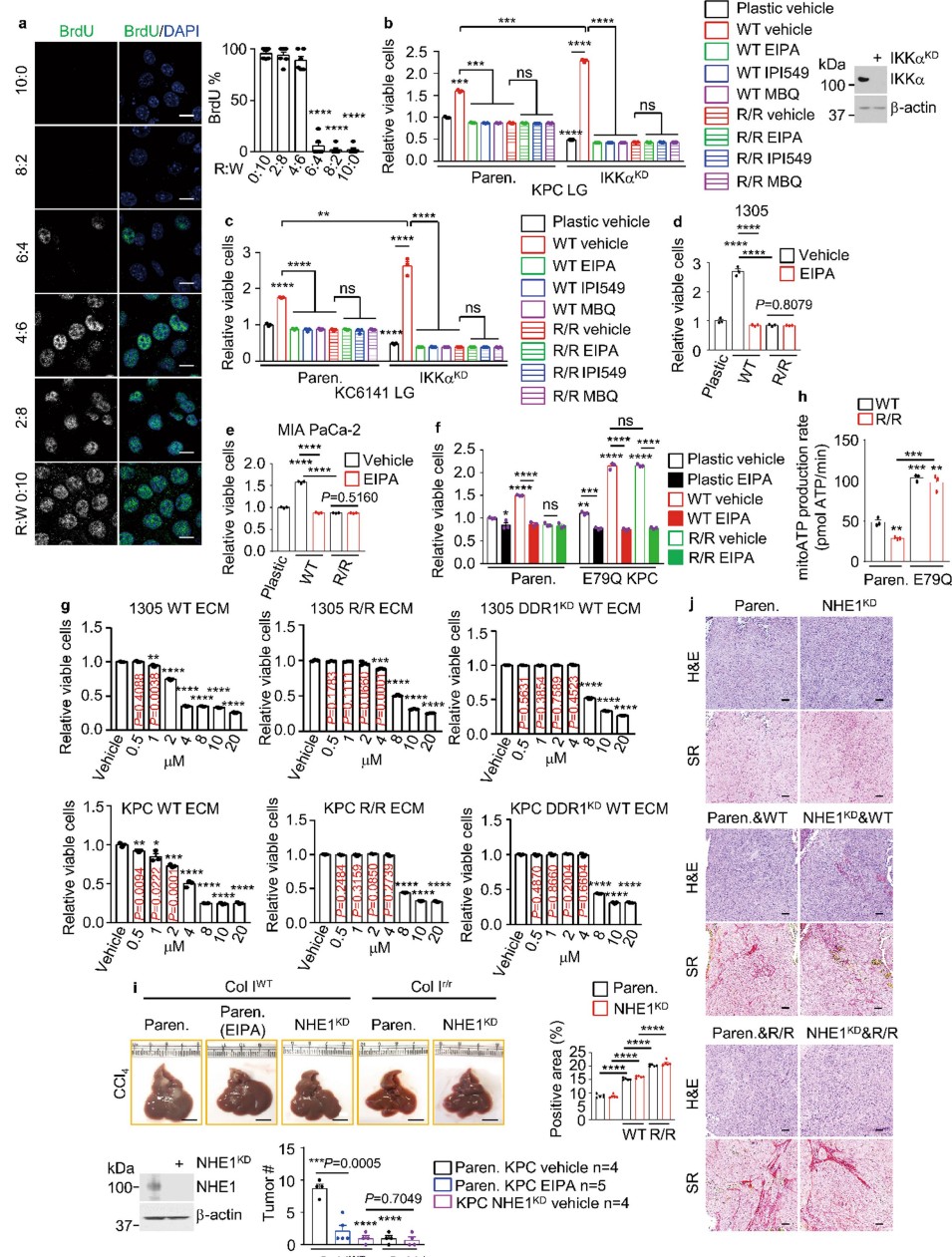

**Extended Data Fig. 8 | Effect of macropinocytosis and mitochondria on Col I-controlled PDAC cell growth. a**, Bromodeoxyuridine (BrdU) incorporation into KPC cells plated on ECM mixtures produced by R/R and WT fibroblasts (R:W). Scale bars, 10 μm. **b,c**, Paren. and IKKα^KD KPC (**b**) or KC (**c**) cells were plated −/+ WT or R/R ECM and incubated in LG −/+ indicated reagents. Viable cells were measured after 3 days and depicted relative to untreated plastic-plated Paren. cells. (**c**). IKKα KD efficiency is shown on the right (**b**). **d,e**, Viable 1305 (**d**) or MIA PaCa-2 (**e**) cells plated as above and incubated in LG −/+ EIPA. **f**, Viable Paren. and E79Q KPC cells plated, treated, and presented as above. **g**, Paren. and DDR1^KD 1305 or KPC cells plated on indicated ECM preparations and incubated in LG medium −/+ indicated tigecycline concentrations for 24 h.

Total viable cells were measured as above and are presented relative to the untreated cells. **h**, Mitochondrial ATP production calculated from Fig. 6b. **i**, Liver morphology and tumour numbers (#) 2 wk after i.s. transplantation of Paren. or NHE1^KD KPC cells into CCl₄-pretreated Col I^WT and Col I^r/r mice −/+ EIPA. NHE1 IB is shown on bottom left. ****P < 0.0001. **j**, H&E and SR staining of s.c. tumours from Fig. 6c. Quantification of the SR positive area is shown on the left. Scale bars, 100 μm. Results in (**a**) (*n* = 6 fields), (**b–g**) (*n* = 3 independent experiments), (**h**) (*n* = 3 per condition), (**j**) (*n* = 5 mice) and (**i**) are mean ± s.e.m. Statistical significance determined by two-tailed *t*-test. Exact *P* values in (**a–c,f,h**) are shown in Source Data.

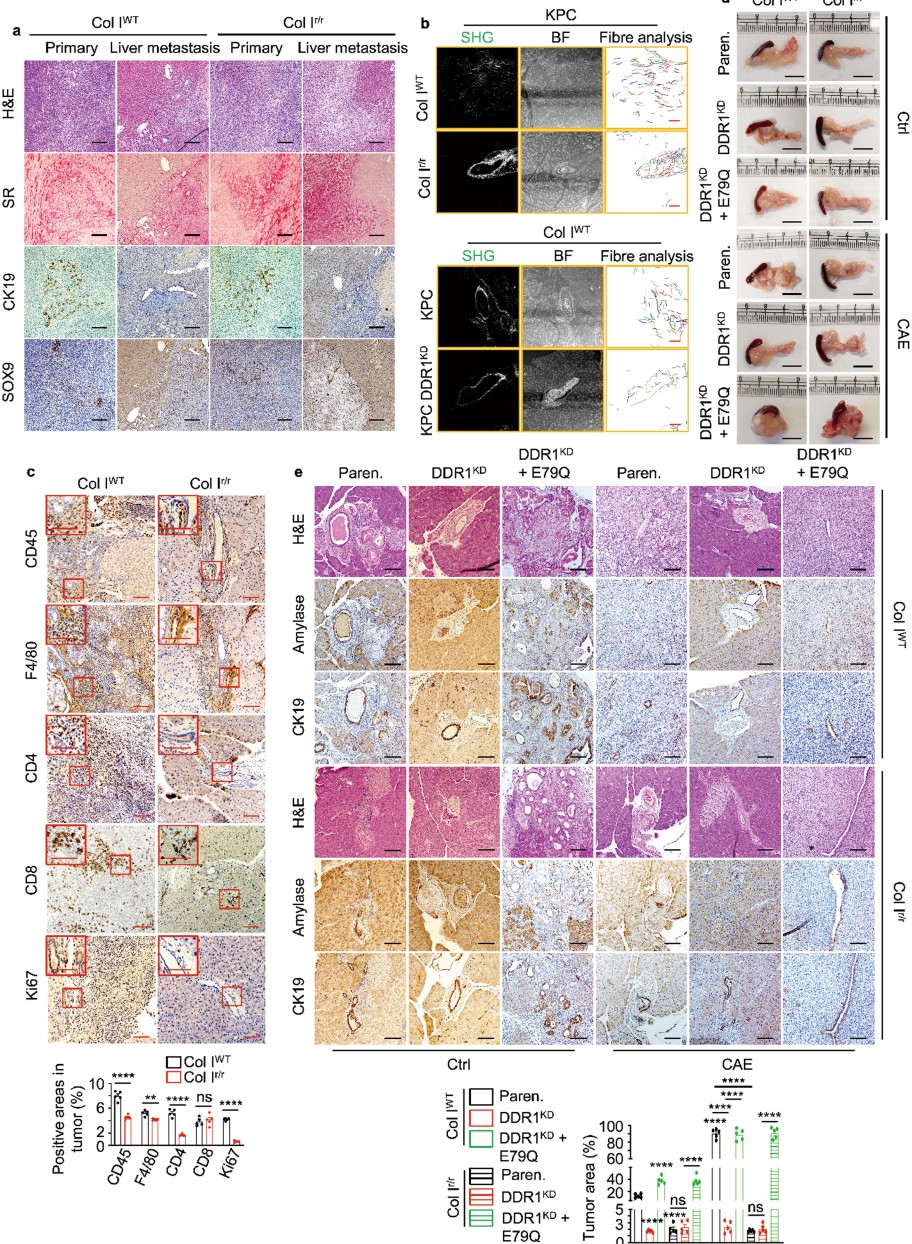

**Extended Data Fig. 9 | The cCol I–DDR1–NRF2 axis controls PDAC growth but not collagen fibre alignment. a**, H&E, SR and cytokeratin 19 (CK19) and SOX9 staining of pancreatic and liver sections from CAE-pretreated Col I$^{WT}$ and Col I$^{r/r}$ mice 4 wk after orthotopic KPC NRF2$^{E79Q}$ cell transplantation. Scale bars, 100 μm. **b**, Tumours formed by orthotopically transplanted Paren. or DDR1$^{KD}$ KPC cells in Col I$^{WT}$ and Col I$^{r/r}$ mice analysed by SHG and collagen fibre individualization. BF-bright field. **c**, IHC of indicated markers in pancreatic sections of Col I$^{WT}$ and Col I$^{r/r}$ mice orthotopically transplanted with KPC cells.

Quantification of staining positivity in tumour areas is shown below. Left to right: ****$P < 0.0001$, **$P = 0.0013$, $P = 0.5351$. **d**, Pancreas morphology 4 wk after orthotopic transplantation of Paren., DDR1$^{KD}$, E79Q/DDR1$^{KD}$ KPC cells into Col I$^{WT}$ and Col I$^{r/r}$ mice pretreated –/+ CAE. Scale bars, 1 cm. **e**, H&E staining and IHC analysis of pancreatic sections from above mice. Quantification of tumour area by ImageJ is shown below. Left to right: ****$P < 0.0001$, $P = 0.5748$, $P = 0.3606$. Results in (**c**,**e**) ($n = 5$ fields) are mean ± s.e.m. Statistical significance determined by two-tailed $t$-test. Scale bars, 100 μm.

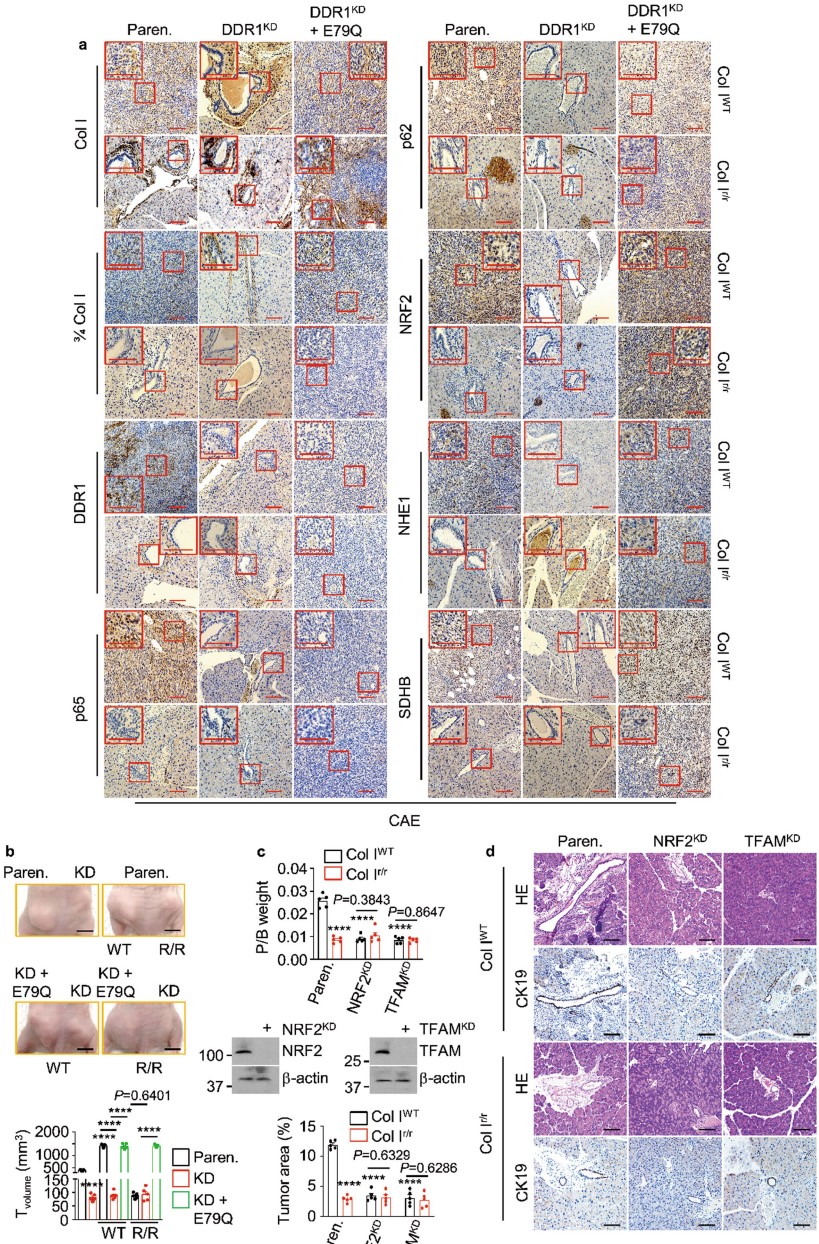

**Extended Data Fig. 10 | cCol I stimulates PDAC growth, macropinocytosis and mitochondrial biogenesis through the DDR1–NRF2 axis. a**, IHC of pancreatic sections of CAE-pretreated Col I$^{WT}$ or Col I$^{r/r}$ mice 4 wk after orthotopic transplantation of Paren., DDR1$^{KD}$ and E79Q/DDR1$^{KD}$ KPC cells. Boxed areas were further magnified. Scale bars, 100 μm. **b**, Representative images and sizes of s.c. tumours generated by Paren., DDR1$^{KD}$ (KD) and E79Q/KD 1305 cells transplanted −/+ WT or R/R fibroblasts into *Nu/Nu* mice. Scale bars, 1 cm. **c**, Pancreas weight relative to body weight (P/B) of Col I$^{WT}$ or Col I$^{r/r}$ mice 4 wk after orthotopic transplantation of Paren., NRF2$^{KD}$ and TFAM$^{KD}$ KPC cells. NRF2 and TFAM KD efficiency is shown below. **d**, H&E staining and CK19 IHC of above pancreata. Image J quantification of tumour area is shown on the left. Scale bars, 100 μm. Results in (**b**,**c**) (*n* = 5 mice), (**d**) (*n* = 5 fields) are mean ± s.e.m. Statistical significance was determined by a two-tailed *t*-test. ****$P$ < 0.0001.

# Reporting Summary

## Statistics

For all statistical analyses, confirm that the following items are present in the figure legend, table legend, main text, or Methods section.

| n/a | Confirmed | |
|---|---|---|
| ☐ | ☒ | The exact sample size (*n*) for each experimental group/condition, given as a discrete number and unit of measurement |
| ☐ | ☒ | A statement on whether measurements were taken from distinct samples or whether the same sample was measured repeatedly |
| ☐ | ☒ | The statistical test(s) used AND whether they are one- or two-sided<br>*Only common tests should be described solely by name; describe more complex techniques in the Methods section.* |
| ☒ | ☐ | A description of all covariates tested |
| ☒ | ☐ | A description of any assumptions or corrections, such as tests of normality and adjustment for multiple comparisons |
| ☐ | ☒ | A full description of the statistical parameters including central tendency (e.g. means) or other basic estimates (e.g. regression coefficient) AND variation (e.g. standard deviation) or associated estimates of uncertainty (e.g. confidence intervals) |
| ☐ | ☒ | For null hypothesis testing, the test statistic (e.g. *F*, *t*, *r*) with confidence intervals, effect sizes, degrees of freedom and *P* value noted<br>*Give P values as exact values whenever suitable.* |
| ☒ | ☐ | For Bayesian analysis, information on the choice of priors and Markov chain Monte Carlo settings |
| ☒ | ☐ | For hierarchical and complex designs, identification of the appropriate level for tests and full reporting of outcomes |
| ☒ | ☐ | Estimates of effect sizes (e.g. Cohen's *d*, Pearson's *r*), indicating how they were calculated |

*Our web collection on statistics for biologists contains articles on many of the points above.*

## Software and code

Policy information about availability of computer code

| Data collection | SoftMax 6.5, Leica Application Suite AF 2.6.0.7266, GEPIA2, AxioVision Rel. 4.5 |
|---|---|
| Data analysis | SoftMax 6.5, Leica Application Suite AF 2.6.0.7266, Graphpad Prism 9, GEPIA2, Image J 1.53k, Fiji Image J 1.53q, R (v4.0.2), Seurat4 (v4.0.5), CT-Fire software (v.2.0 beta), HOMER (v4.11), GSEA (4.0.3) |

For manuscripts utilizing custom algorithms or software that are central to the research but not yet described in published literature, software must be made available to editors and reviewers. We strongly encourage code deposition in a community repository (e.g. GitHub). See the Nature Portfolio guidelines for submitting code & software for further information.

## Data

Policy information about availability of data

All manuscripts must include a data availability statement. This statement should provide the following information, where applicable:
- Accession codes, unique identifiers, or web links for publicly available datasets
- A description of any restrictions on data availability
- For clinical datasets or third party data, please ensure that the statement adheres to our policy

RNA-seq data for KPC960 cells grown on WT or R/R ECM have been deposited in the Gene Expression Omnibus under the accession code GSE206218, scRNA-seq dataset for 5 primary PDAC tumors and 1 PDAC liver metastasis were obtained from the published GEO dataset (GSE156405) (Lee et al., Clinical Cancer Research, 2021).Custom computer code used in above scRNA-seq analysis is available at https://github.com/ajynair/Collagen_DDR1_PDACmets.The mouse reference genome

GRCm38 was obtained from Ensembl version 86 or 102 (http://nov2020.archive.ensembl.org/Mus_musculus/Info/Index). TCGA data are available within GEPIA2 (http://gepia2.cancer-pku.cn/#dataset).

# Human research participants

Policy information about studies involving human research participants and Sex and Gender in Research.

| | |
|---|---|
| Reporting on sex and gender | no sex- and gender-based analyses have been performed. This study did not involve sex or gender research. |
| Population characteristics | Human PDAC specimens were acquired from patients who were diagnosed with PDAC between January 2017 and May 2021 at The Affiliated Drum Tower Hospital of Nanjing University Medical School (Nanjing, Jiangsu, China). All patients received standard surgical resection and did not receive chemotherapy before surgery. Paraffin embedded tissues were processed by a pathologist after surgical resection and confirmed as PDAC prior to further investigation. Overall survival duration was defined as the time from date of diagnosis to that of death or last known follow-up examination. Survival information was available for 81 of the 106 patients. Patient number and age in the parentheses are listed as follows, |

#21-41(55)
#21-42(57)
#21-43(58)
#21-44(52)
#21-45(52)
#21-46(54)
#21-47(57)
#21-48(58)
#21-49(53)
#21-50(68)
#21-51(40)
#21-52(57)
#21-53(51)
#21-54(53)
#21-55(55)
#21-56(63)
#21-57(64)
#21-58(57)
#21-59(55)
#21-60(58)
#19-11(69)
#19-12(65)
#19-14(62)
#19-16(68)
#19-20(61)
#19-21(73)
#19-22(66)
#19-23(71)
#19-24(63)
#19-25(74)
#19-26(72)
#19-27(74)
#19-28(52)
#19-29(56)
#19-30(60)
#19-31(53)
#19-32(62)
#19-33(70)
#19-34(54)
#19-35(68)
#19-37(49)
#19-39(81)
#19-40(67)
#19-43(53)
#19-44(57)
#19-45(76)
#19-46(62)
#19-47(57)
#19-48(71)
#19-49(70)
#19-50(49)
#19-52(57)
#19-54(70)
#19-55(73)
#19-56(65)
#19-57(77)
#19-58(62)

```
#19-60(49)
#19-61(77)
#19-62(41)
#19-63(69)
#19-64(62)
#19-65(64)
#19-66(63)
#19-67(71)
#19-68(64)
#19-69(67)
#19-70(67)
#19-71(61)
#19-72(66)
#19-73(61)
#19-74(64)
#19-75(73)
#19-76(66)
#19-77(58)
#19-78(67)
#19-79(65)
#19-80(76)
#19-81(81)
#19-82(57)
#19-83(64)
#19-84(50)
#19-85(50)
#19-86(54)
#19-87(75)
#19-88(82)
#19-89(55)
#19-90(71)
#19-91(61)
#19-92(75)
#19-93(64)
#19-95(48)
#19-97(49)
#19-100(67)
#19-101(61)
#19-103(78)
#19-104(85)
#19-105(55)
#19-107(69)
#19-108(87)
#19-109(67)
#19-110(75)
#19-111(58)
#19-112(44)
#19-113(58)
#19-115(57)
#19-116(57)
```

| | |
|---|---|
| Recruitment | Human PDAC specimens were acquired from patients who were diagnosed with PDAC between January 2017 and May 2021 at The Affiliated Drum Tower Hospital of Nanjing University Medical School (Nanjing, Jiangsu, China). All patients received standard surgical resection and did not receive chemotherapy before surgery. Paraffin embedded tissues were processed by a pathologist after surgical resection and confirmed as PDAC prior to further investigation. Informed consent for tissue analysis was obtained before surgery. |
| Ethics oversight | The study was approved by the Institutional Ethics Committee of The Affiliated Drum Tower Hospital with IRB #2021-608-01. |

Note that full information on the approval of the study protocol must also be provided in the manuscript.

# Field-specific reporting

Please select the one below that is the best fit for your research. If you are not sure, read the appropriate sections before making your selection.

☒ Life sciences    ☐ Behavioural & social sciences    ☐ Ecological, evolutionary & environmental sciences

For a reference copy of the document with all sections, see nature.com/documents/nr-reporting-summary-flat.pdf

# Life sciences study design

All studies must disclose on these points even when the disclosure is negative.

| | |
|---|---|
| Sample size | No statistical methods were used to predetermine samples sizes for in vitro experiments. Sample sizes were chosen in order to be able to perform statistical analyses, as is standard in the field and based on previous studies (Su et al., Cancer Cell, 2021; Zhong et al., Nature, 2018). For in vivo experiments, based on their genotypes, gender- and age matched mice were randomly allocated to experimental groups. Because our mice were inbred and age- and gender-matched, similar variance was assumed between different experimental groups. No sample size pre-estimation was performed but we used as many mice per group as possible to minimize type I/II errors. |
| Data exclusions | No data were excluded for all the analyses described. |
| Replication | All the experiments except IHC analysis of 106 patient samples were repeated for at least three times. However, all the antibodies used in IHC analysis of 106 patient samples were confirmed their specificity in several patient samples at least three times. Statistical analysis were done to ensure significance. All attempts of replication were successful. |
| Randomization | Age- , gender-, and equal average tumor volumes-matched mice were randomly allocated to different experimental groups based on their genotypes. For experiments other than mice, we did not carry out randomization because it's either irrelevant or not applicable to these studies. |
| Blinding | Investigators were not blinded to the group allocations except for microscopic analysis of immunofluorescent or IHC staining results. For other experiments, the investigators were not blinded since analyses relied on unbiased measurements of quantitative parameters. Standardized procedures for data collection and analysis were used to prevent bias. |

# Reporting for specific materials, systems and methods

We require information from authors about some types of materials, experimental systems and methods used in many studies. Here, indicate whether each material, system or method listed is relevant to your study. If you are not sure if a list item applies to your research, read the appropriate section before selecting a response.

## Materials & experimental systems

| n/a | Involved in the study |
|---|---|
| ☐ | ☒ Antibodies |
| ☐ | ☒ Eukaryotic cell lines |
| ☒ | ☐ Palaeontology and archaeology |
| ☐ | ☒ Animals and other organisms |
| ☒ | ☐ Clinical data |
| ☒ | ☐ Dual use research of concern |

## Methods

| n/a | Involved in the study |
|---|---|
| ☒ | ☐ ChIP-seq |
| ☒ | ☐ Flow cytometry |
| ☒ | ☐ MRI-based neuroimaging |

## Antibodies

| | |
|---|---|
| Antibodies used | Guinea pig anti-p62 polyclonal antibody (GP62-C, Progen), rabbit anti-NRF2 polyclonal antibody (ABclonal, A11159), rabbit anti-COL1A1 monoclonal antibody (CST, 72026, E8F4L), mouse anti-COL1A1 monoclonal antibody (Santa Cruz, sc-293182, 3G3), rabbit anti-3/4 COL1A1 polyclonal antibody (Immunoglobe, 0217-050), mouse anti-TIM23 monoclonal antibody (Santa Cruz, sc-514463, H-8), rabbit anti-phospho-DDR1 (pTyr513) polyclonal antibody (Sigma, SAB4504671), mouse anti-DDR1 monoclonal antibody (Santa Cruz, sc-390268, D-10), rabbit anti-KEAP1 monoclonal antibody (CST, 8047, D6B12), rabbit anti-NF-κB p65 monoclonal antibody (CST, 8242, D14E12), rabbit anti-Histone H3 polyclonal antibody (ABclonal, A2348), rat anti-CD326 (EpCAM) monoclonal antibody (ThermoFisher, 13-5791-80, G8.8), mouse anti-IKKa monoclonal antibody (Invitrogen, MA5-16157, 14A231), mouse anti-Actin monoclonal antibody (Sigma, A4700, AC-40), rabbit anti-GFP polyclonal antibody (ThermoFisher, A-11122), chicken anti-GFP/YFP/CFP polyclonal antibody (Abcam ab13970), mouse anti-Flag monoclonal antibody (Sigma, F3165, M2), rabbit anti-Flag polyclonal antibody (Sigma, F7425), rabbit anti-TFAM polyclonal antibody (Abcam, ab131607), rabbit anti-PGC1 polyclonal antibody (Sigma, ABE868), rabbit anti-Phospho-AMPKα (Thr172) monoclonal antibody (CST, 2535, 40H9), rabbit anti-AMPKα monoclonal antibody (CST, 5832, D63G4), mouse anti-6X His tag monoclonal antibody (Abcam, ab18184, HIS.H8), rabbit anti-E-Cadherin monoclonal antibody (CST, 3195, 24E10), rabbit anti-CD138/SDC1 polyclonal antibody (ThermoFisher, 36-2900), mouse anti-NHE-1 monoclonal antibody (Santa Cruz, sc-136239, 54), rabbit anti-PI3 Kinase p110γ monoclonal antibody (CST, 5405, D55D5), mouse anti-ATP5A monoclonal antibody (Santa Cruz, sc-136178, 51), mouse anti-ATP5B monoclonal antibody (Sigma, MAB3494, 4.3E8.D1), mouse anti-UQCRC2 monoclonal antibody (Santa Cruz, sc-390378, G-10), mouse anti-SDHB monoclonal antibody (Santa Cruz, sc-271548, G-10), rabbit anti-SDHB monoclonal antibody (CST, 92649, E3H9Z), mouse anti-NDUFB7 monoclonal antibody (Santa Cruz, sc-365552, F-8), rabbit anti-COX1/MT-CO1 polyclonal antibody (CST, 62101), rabbit anti- SMA polyclonal antibody (Abcam, ab5694), rabbit anti-MMP1 monoclonal antibody (Abcam, ab52631, EP1247Y), rabbit anti-Ki67 monoclonal antibody (GeneTex, GTX16667, SP6), rabbit anti-CDC42 polyclonal antibody (ThermoFisher, PA1-092), mouse anti-HSP90 monoclonal antibody (Santa Cruz, sc-13119, F-8), rabbit anti-α-Amylase polyclonal antibody (Sigma, A8273), goat anti-cytokeratin 19 polyclonal antibody (Santa Cruz, sc-33111), rabbit anti-SOX9 polyclonal antibody (Santa Cruz, sc-20095), mouse anti-cytokeratin 18 polyclonal antibody (GeneTex, GTX105624), rabbit anti-LAIR1 polyclonal antibody (ThermoFisher, H00003903-D01P), mouse anti-Endo180/MRC2 monoclonal antibody (Santa Cruz, sc-271148, B-10), mouse anti-Integrin β1/ITGB1 monoclonal antibody (Santa Cruz, sc-374429, A-4), Rat anti-CD45 monoclonal antibody (ThermoFisher, 14-0451-85, 30-F11), Mouse anti-CD68 monoclonal antibody (ThermoFisher, MA5-13324, KP1), Rabbit anti-CD163 monoclonal |

antibody (Abcam, ab182422, EPR19518), Rat anti-F4/80 monoclonal antibody (ThermoFisher, MF48000, BM8), Rabbit anti-CD4 monoclonal antibody (Abcam, ab183685, EPR19514), Rabbit anti-Ki67 polyclonal  antibody (Abcam, ab15580), Rabbit anti-CD8 monoclonal antibody (Abcam, ab217344, EPR21769), HRP goat anti-chicken IgY antibody (Santa Cruz, sc-2428), HRP goat anti-rabbit IgG antibody (CST, 7074), HRP horse anti-mouse IgG antibody (CST, 7076), HRP streptavidin (Pharmingen, 554066), Biotin goat anti-mouse IgG (Pharmingen, 553999), Biotin goat anti-rabbit IgG (Pharmingen, 550338), Biotin mouse anti-goat IgG (Santa Cruz, sc-2489). Alexa 594-, Alexa 647-, and Alexa 488-conjugated secondary antibodies were used: donkey anti-mouse IgG, donkey anti-rabbit IgG, goat anti-chicken IgY (Molecular Probes, Invitrogen).

**Validation**

All the following antibodies have been validated according to manufacturer's manuals and re-validated by immunoblot (IB), or immunofluorescence staining (IF) or immunohistochemistry (IHC) results from this manuscript:

Guinea pig anti-p62 polyclonal antibody (GP62-C, Progen) (IB, human and mouse): https://us.progen.com/anti-p62-SQSTM1-C-terminus-guinea-pig-polyclonal-serum/GP62-C

rabbit anti-NRF2 polyclonal antibody (ABclonal, A11159) (IB, IHC, IF, human and mouse): https://abclonal.com/catalog-antibodies/NRF2RabbitpAb/A11159

rabbit anti-COL1A1 monoclonal antibody (CST, 72026) (IB, human and mouse): https://www.cellsignal.com/products/primary-antibodies/col1a1-e8f4l-xp-rabbit-mab/72026

mouse anti-COL1A1 monoclonal antibody (Santa Cruz, sc-293182) (IB, human and mouse): https://www.scbt.com/p/col1a1-antibody-3g3

rabbit anti-3/4 COL1A1 polyclonal antibody (Immunoglobe, 0217-050) (IB, IF, IHC, human and mouse): https://www.immunoglobe.com/antibodies/items/collagen_cleavage_site.html

mouse anti-TIM23 monoclonal antibody (Santa Cruz, sc-514463) (IF, human and mouse): https://www.scbt.com/p/tim23-antibody-h-8

rabbit anti-phospho-DDR1 (pTyr513) polyclonal antibody (Sigma, SAB4504671) (IB, human and mouse): https://www.sigmaaldrich.com/US/en/product/sigma/sab4504671

mouse anti-DDR1 monoclonal antibody (Santa Cruz, sc-390268) (IB, human and mouse): https://www.scbt.com/p/ddr1-antibody-d-10

rabbit anti-KEAP1 monoclonal antibody (CST, 8047) (IB, human and mouse): https://www.cellsignal.com/products/primary-antibodies/keap1-d6b12-rabbit-mab/8047

rabbit anti-NF-κB p65 monoclonal antibody (CST, 8242) (IB, human and mouse): https://www.cellsignal.com/products/primary-antibodies/nf-kb-p65-d14e12-xp-rabbit-mab/8242

rabbit anti-Histone H3 polyclonal antibody (ABclonal, A2348) (IB, human and mouse): https://abclonal.com/catalog-antibodies/HistoneH3RabbitpAb/A2348

rat anti-CD326 (EpCAM) monoclonal antibody (ThermoFisher, 13-5791-80) (IF, IHC, mouse): https://www.thermofisher.com/antibody/product/CD326-EpCAM-Antibody-clone-G8-8-Monoclonal/13-5791-80

mouse anti-IKKa monoclonal antibody (Invitrogen, MA5-16157) (IB, human and mouse): https://www.thermofisher.com/antibody/product/IKK-alpha-Antibody-clone-14A231-Monoclonal/MA5-16157

mouse anti-Actin monoclonal antibody (Sigma, A4700) (IB, human and mouse): https://www.sigmaaldrich.com/US/en/product/sigma/a4700

rabbit anti-GFP polyclonal antibody (ThermoFisher, A-11122) (IB):https://www.thermofisher.com/antibody/product/GFP-Antibody-Polyclonal/A-11122

chicken anti-GFP/YFP/CFP polyclonal antibody (Abcam ab13970) (IF): https://www.abcam.com/gfp-antibody-ab13970.html

mouse anti-Flag monoclonal antibody (Sigma, F3165) (IB, human and mouse): https://www.sigmaaldrich.com/US/en/product/sigma/f3165

rabbit anti-Flag polyclonal antibody (Sigma, F7425) (IB, human and mouse): https://www.sigmaaldrich.com/US/en/product/sigma/f7425

rabbit anti-TFAM polyclonal antibody (Abcam, ab131607) (IB, mouse ): https://www.citeab.com/antibodies/754337-ab131607-anti-mttfa-antibody-mitochondrial-marker

rabbit anti-PGC1a polyclonal antibody (Sigma, ABE868) (IB, human and mouse): https://www.sigmaaldrich.com/US/en/product/mm/abe868

rabbit anti-Phospho-AMPKα (Thr172) monoclonal antibody (CST, 2535) (IB, human and mouse): https://www.cellsignal.com/products/primary-antibodies/phospho-ampka-thr172-40h9-rabbit-mab/2535

rabbit anti-AMPKα monoclonal antibody (CST, 5832) (IB, human and mouse): https://www.cellsignal.com/products/primary-antibodies/ampka-d63g4-rabbit-mab/5832

mouse anti-6X His tag monoclonal antibody (Abcam, ab18184) (IB, mouse): https://www.abcam.com/6x-his-tag-antibody-hish8-ab18184.html

rabbit anti-E-Cadherin monoclonal antibody (CST, 3195) (IF, human and mouse): https://www.cellsignal.com/products/primary-antibodies/e-cadherin-24e10-rabbit-mab/3195

rabbit anti-CD138/SDC1 antibody (ThermoFisher, 36-2900) (IB, human and mouse): https://www.thermofisher.com/antibody/product/CD138-Antibody-Polyclonal/36-2900

mouse anti-NHE-1 monoclonal antibody (Santa Cruz, sc-136239) (IB, human and mouse): https://www.scbt.com/p/nhe-1-antibody-54

rabbit anti-PI3 Kinase p110γ monoclonal antibody (CST, 5405) (IB, human and mouse): https://www.cellsignal.com/products/primary-antibodies/pi3-kinase-p110g-d55d5-rabbit-mab/5405

mouse anti-ATP5A monoclonal antibody (Santa Cruz, sc-136178) (IB, human and mouse): https://www.scbt.com/p/atp5a-antibody-51

mouse anti-ATP5B monoclonal antibody (Sigma, MAB3494) (IB, human and mouse): https://www.sigmaaldrich.com/deepweb/assets/sigmaaldrich/product/documents/309/124/mab3494.pdf

mouse anti-UQCRC2 monoclonal antibody (Santa Cruz, sc-390378) (IB, human and mouse): https://www.scbt.com/p/uqcrc2-antibody-g-10

mouse anti-SDHB monoclonal antibody (Santa Cruz, sc-271548) (IB, human and mouse): https://www.scbt.com/p/sdhb-antibody-g-10

rabbit anti-SDHB monoclonal antibody (CST, 92649) (IHC, human and mouse): https://www.cellsignal.com/products/primary-antibodies/sdhb-e3h9z-xp-rabbit-mab/92649

mouse anti-NDUFB7 monoclonal antibody (Santa Cruz, sc-365552) (IB, human and mouse): https://www.scbt.com/p/ndufb7-antibody-f-8

rabbit anti-COX1/MT-CO1 polyclonal antibody (CST, 62101) (IB, human and mouse): https://www.cellsignal.com/products/primary-antibodies/cox1-mt-co1-antibody/62101

rabbit anti-aSMA polyclonal antibody (Abcam, ab5694) (IHC, mouse and human): https://www.abcam.com/alpha-smooth-muscle-actin-antibody-ab5694.html
rabbit anti-MMP1 monoclonal antibody (Abcam, ab52631) (IHC, human): https://www.abcam.com/mmp1-antibody-ep1247y-ab52631.html
rabbit anti-Ki67 monoclonal antibody (GeneTex, GTX16667) (IHC, human and mouse): https://www.genetex.com/Product/Detail/Ki67-antibody-SP6/GTX16667
rabbit anti-CDC42 polyclonal antibody (ThermoFisher, PA1-092) (IHC, IB, human and mouse): https://www.thermofisher.com/antibody/product/Cdc42-Antibody-Polyclonal/PA1-092
mouse anti-HSP90 monoclonal antibody (Santa Cruz, sc-13119) (IB, human and mouse): https://www.scbt.com/p/hsp-90alpha-beta-antibody-f-8
rabbit anti-α-Amylase polyclonal antibody (Sigma, A8273) (IHC, human): https://www.sigmaaldrich.com/US/en/product/sigma/a8273
mouse anti-cytokeratin 18 monoclonal antibody (GeneTex, GTX105624) (IF, human and mouse): https://www.genetex.com/Product/Detail/Cytokeratin-18-antibody-N2C2-Internal/GTX105624
rabbit anti-LAIR1 polyclonal antibody (ThermoFisher, H00003903-D01P) (IB, human and mouse): https://www.thermofisher.com/antibody/product/LAIR1-Antibody-Polyclonal/H00003903-D01P
mouse anti-Endo180/MRC2 monoclonal antibody (Santa Cruz, sc-271148) (IB, human): https://www.scbt.com/p/endo180-antibody-b-10
mouse anti-Integrin β1/ITGB1 antibody (Santa Cruz, sc-374429) (IB, human and mouse): https://www.scbt.com/p/integrin-beta1-antibody-a-4
Rat anti-CD45 antibody (ThermoFisher, 14-0451-85) (IHC, mouse): https://www.thermofisher.com/antibody/product/CD45-Antibody-clone-30-F11-Monoclonal/14-0451-85
Mouse anti-CD68 antibody (ThermoFisher, MA5-13324) (IHC, human): https://www.thermofisher.com/antibody/product/CD68-Antibody-clone-KP1-Monoclonal/MA5-13324
Rabbit anti-CD163 antibody (Abcam, ab182422) (IHC, human and mouse): https://www.abcam.com/cd163-antibody-epr19518-ab182422.html
Rat anti-F4/80 antibody (ThermoFisher, MF48000) (IHC, mouse): https://www.thermofisher.com/antibody/product/F4-80-Antibody-clone-BM8-Monoclonal/MF48000
Rabbit anti-CD4 antibody (Abcam, ab183685) (IHC, mouse): https://www.abcam.com/cd4-antibody-epr19514-ab183685.html
Rabbit anti-Ki67 antibody (Abcam, ab15580) (IHC, mouse and human): https://www.abcam.com/ki67-antibody-ab15580.html
Rabbit anti-CD8 antibody (Abcam, ab217344) (IHC, mouse): https://www.abcam.com/cd8-alpha-antibody-epr21769-ab217344.html
HRP goat anti-chicken IgY antibody (Santa Cruz, sc-2428) (IF): https://datasheets.scbt.com/sc-2428.pdf
HRP goat anti-rabbit IgG antibody (CST, 7074) (IB): https://www.cellsignal.com/products/secondary-antibodies/anti-rabbit-igg-hrp-linked-antibody/7074
HRP horse anti-mouse IgG antibody (CST, 7076) (IB): https://www.cellsignal.com/products/secondary-antibodies/anti-mouse-igg-hrp-linked-antibody/7076
HRP streptavidin (Pharmingen, 554066) (IHC): https://www.bdbiosciences.com/content/dam/bdb/products/global/reagents/immunoassay-reagents/elisa/554066_base/pdf/554066.pdf
Biotin goat anti-mouse IgG (Pharmingen, 553999) (IHC, mouse): https://www.bdbiosciences.com/content/dam/bdb/products/global/reagents/flow-cytometry-reagents/research-reagents/single-color-antibodies-ruo/553999_base/pdf/553999.pdf
Biotin goat anti-rabbit IgG (Pharmingen, 550338) (IHC): https://www.bdbiosciences.com/content/dam/bdb/products/global/reagents/flow-cytometry-reagents/research-reagents/single-color-antibodies-ruo/550338_base/pdf/550338.pdf
Biotin mouse anti-goat IgG (Santa Cruz, sc-2489) (IHC): https://www.scbt.com/p/mouse-anti-goat-igg-b
Alexa 594-, Alexa 647-, and Alexa 488-conjugated secondary antibodies were used: donkey anti-mouse IgG, donkey anti-rabbit IgG, goat anti-chicken IgY (Molecular Probes, Invitrogen) (IF): https://www.thermofisher.com/us/en/home/life-science/antibodies/secondary-antibodies/fluorescent-secondary-antibodies/alexa-fluor-plus-secondary-antibodies.html

# Eukaryotic cell lines

Policy information about cell lines and Sex and Gender in Research

| Cell line source(s) | MIA PaCa-2 (CRL-1420; RRID: CVCL_0428) cell was obtained from ATCC. UN-KC-6141 (RRID: CVCL_1U11) and UN-KPC-960 (RRID: CVCL_1U12) were obtained from Surinder K. Batra. WT and R/R fibroblasts were generated at Dr. David Brenner lab. 1305 primary human PDAC cells were generated by Dr. Andrew M. Lowy lab from a human PDAC PDX. |
|---|---|
| Authentication | MIA Paca-2 has been authenticated by ATCC and UN-KC-6141 and UN-KPC-960 have been authenticated by Surinder K. Batra lab before delivery to our lab. And cell lines are routinely authenticated in-house by cell morphology. |
| Mycoplasma contamination | All cell lines are routinely tested negative for mycoplasma contamination. |
| Commonly misidentified lines (See ICLAC register) | No commonly misidentified cell lines were used. |

# Animals and other research organisms

Policy information about studies involving animals; ARRIVE guidelines recommended for reporting animal research, and Sex and Gender in Research

| Laboratory animals | Female homozygous Nu/Nu nude mice and C57BL/6 mice were obtained at 6 weeks of age from Charles River Laboratories and The Jackson Laboratory, respectively. 3-month-old Col1a1+/+ (Col lWT) or Col1a1r/r (Col lr/r) mice on a C57BL/6 background obtained from Dr. David Brenner at UCSD were used in this study (indicated in the Methods) and and were previously described. Age- and sex-matched (except where otherwise indicated) male and female mice of each genotype were generated as littermates for use in |
|---|---|

experiments in which different genotypes were compared. All mice were maintained in filter-topped cages on autoclaved food and water, and experiments were performed in accordance with UCSD Institutional Animal Care and Use Committee and NIH guidelines and regulations on age and gender-matched littermates. Dr. Karin's Animal Protocol S00218 was approved by the UCSD Institutional Animal Care and Use Committee. Mice were housed in well filter-topped cages in constant temperature, humidity and pathogen-free controlled environment (23°C ± 2°C, 50-60%), with a standard 12 h light/ 12 h dark cycle, plenty of water and food in their cages, which were described in Methods section.

Wild animals

The study did not involve wild animals.

Reporting on sex

This study did not involve sex research. But Sex-matched male and female mice of each genotype were generated as littermates for use in experiments in which different genotypes were compared.

Field-collected samples

The study did not involve field-collected samples.

Ethics oversight

Dr. Karin's Animal Protocol S00218 was approved by the UCSD Institutional Animal Care and Use Committee

Note that full information on the approval of the study protocol must also be provided in the manuscript.

