## [Peer Review File · Nature]

Redactions – unpublished data

Manuscript Title: Collagenolysis dependent DDR1 signaling dictates pancreatic cancer outcome

Reviewer Comments & Author Rebuttals

Reviewer Reports on the Initial Version:

Hua Su and colleagues describe in this manuscript entitled “Collagen remodeling dictates pancreatic cancer bioenergetics and outcome through DDR1 activation and degradation” a novel role of collagen signaling in pancreatic cancer. The authors describe distinct roles of MMP1 cleaved collagen I or full length, intact collagen I in DDR1 biology, activation and down-stream signaling, affecting cancer development. Whereas cleaved Col I promotes PDAC growth through DDR1 signaling (involving NFKb/p62 and NRF2) full length, intact Col I induces DDR1 degradation and restrains PDAC growth. Using different mouse models, human tissue and big human cohorts, the authors underline their hypotheses experimentally and show clinical relevance.

Moreover, the included, well characterized patient cohorts to find new stratification criteria for the prognosis of improved median survival are very impressive. This is a highly innovative and clinically relevant study which opens a new field in pancreatic cancer. The study indeed presents a novel pathway in PDAC, opening up several new opportunities for therapy. Conceptually this finding could also be of importance in other cancers with high stromal load.

On the whole the experiments are well executed, well controlled, engaging excellent mouse models and analyses - appropriately chosen and state of the art. This manuscript is very well written and has a red line – enabling the reader to follow the concepts and ideas of the authors in a structure and “good to follow” manner. The most important message of this paper – that Col I remodeling is prognostic for PDAC patient survival – is indeed a big step forward in understanding PDAC pathology in patients – opening novel therapeutic approaches. This referee has a couple of questions/points which might further enhance the clarity of this outstanding manuscript.

Figure 1:

Figure 1a: these are fascinating data, clearly underlining the central hypothesis of this manuscript. One aspect that would be interesting. Does this the level of cCol I correlate in any way with an altered inflammatory signature, which anyhow in PDAC is special and rather a low profile. E.g. in the context of cColl and enhanced NFKb expression would this lead to any type of alterations?
Figure 1b-e: Very impressive in vivo data strongly supporting the hypothesis of the authors.
Figure 1 b, c,d,e: Please add scale bars.

Figure 2:

Figure 2b: The single cell analysis data on the expression of DDR1 is highly informative, maybe should be more focused on 1-2 receptors and enlarged. The rest could be shown in the supplemental material.

Figure 2c: did the authors also check whether other NFKb-related pathways are on besides canonical NFKb (nuclear RelA). What about non-canonical NFKb signaling, as well as other related pathways? kDa is missing

Figure 2k: Results on Col I controlling MP and mitochondrial biogenesis through the DDR1-NF- κ B-p62-NRF2 cascade are convincing! Just to analyze this in a bit more exact fashion it would be helpful to densitometrically analyze the shown data – at least for the key components DDR1 NFKb p62 – maybe even in a correlation analysis.

Also here an analysis of possible changes in the most important innate and adaptive immune cells would be very helpful. Any changes in cell survival, or cell death?

Figure 2l: Please add kDa.

Figure 2m: Colocalization of GFP-DDR1 and poly-ubiquitin (poly-ub) in GFP-DDR1 expressing 1305 cells cocultured with WT or R/R fibroblasts is very convincing!

Figure 3:

Data in figure 3 underline the clinical relevance of this study. Stains appear very specific and have a low background noise.

Figure 3c: it would be helpful if the authors could show one more blot comparing cCol I low DDR high as well as c Col I low NRF2 hi. In the main Figure.

Data are very convincing demonstrating that the identified pathway indeed is operational in human pathogenesis dictating patient survival.

Figure 4:

Data in Figure 4 are highly convincing, underlining that this pathway is therapeutically targetable. Importantly, experiments on NFKb inhibition (IKKb) or inhibition of mitochondrial protein synthesis causing decreased tumor growth are very impressive.

Minor; Scale bars in all macroscopic images are missing. kDa are missing in Western blots.

On the whole this is an impressive, innovative manuscript with a very focused and well supported message, with extremely high potential for clinical applications to treat or prognostically evaluate PDAC.

Referee #2 (Remarks to the Author):

Karin and colleagues investigated the role of ECM collagen in the progression of PDAC. PDAC showed a lot of ECM deposition and it may not be only the consequence of the disease. ECM, mainly collagen, in PDAC may also have some functional effects on PDAC growth. Until now, the results are controversial, some reports said collagen promotes, but others reported collagen restricts PDAC growth. This study tried to address this very fundamental but very important question. Therefore,

although the study concept is simple, the scientifically, this study is very important. The research team found that MMP-cleaved collagen can promote, but intact collagen has opposite function on PDAC growth. Cleaved collagen activates NF- κ B through the receptor DDR1. Furthermore, they found collagen-induced NF- κ B activation affects the p62-NRF2 signaling to promote PDAC growth by modulating mitochondrial biogenesis. Overall, this study is significant and could lead to a paradigm shift in the field of cancer research. This reviewer has several comments that should be addressed.

1. In Fig.1a, what assay was used for the evaluation of cCol in PDAC patients, either Immunoblots or immunostaining? It is unclear from the method description. This reviewer assumes this is immunostaining when looked at Figure 3.
2. The authors used MMP1,2,8,9,13,14 as a collagen cleavage signature. LOX genes are involved in collagen cross link. Is LOX gene signature also associated with patient outcomes?
3. Regarding Extended Data Fig. 1f, the text (line 100) mentioned CCl4 pretreatment, but the Figure and legend mentioned IKKa KD mice. Please clarify.
4. In Fig.1i and Extended Fig. 2f,h, it is unclear why 3H uptake and ATP production are high in the ColDelta/Colr/r condition (green bar, vehicle) even no cleaved collagen production (Fig.1i WB). This reviewer thought cleaved collagen is required for the source of 3H uptake and subsequent ATP production. Both black bar and purple bar have cleaved collagen, but not green bar. To this reviewer and to the authors' conclusion (line 143-144), these data are inconsistent. Please clarify.
5. In primary PDAC, DDR1 expression seems low in primary PDAC (Fig2b). In contrast, ITGB1 expression is high in both primary and metastatic PDAC (Extended Fig 4b). Because the functional test showed DDR1 is more important than ITGB1, these scRNA-seq data seems not nice. Because the sample number is not many, either removing, adding more sample number, or other staining approaches (immunostaining or RNA scope) is recommended.
6. Line 196-197, if the authors want to claim "..., with other collagens in Col delta fibroblasts acting as ligands". It could be required to use collagenase to degrade all collagens, and then the study needs to show that effect can be disappeared. Depending on the result, there is a possibility that DDR1 ligands, other than collagens, might stimulate the signaling though the impact of this study does not change.

Referee #3 (Remarks to the Author):

Many cancer types are characterized by a fibrotic phenotype with abundant collagen deposition. However, in the field there is inconsistent data regarding the function of collagen in tumor progression. The current study demonstrates that a tumor-promoting function is mediated by cleaved collagen (cCol). cCol enhances bioenergetics and macropinocytosis in pancreatic cancer through DDR1, a RTK that binds triple helical collagen. In contrast the authors show that intact collagen (iCol) promotes DDR1 degradation and has an inhibitory effect on cancer cells. This study provides some clarity on the function and importance of signaling initiated by cleaved collagen in pancreatic cancer progression. Further the study demonstrates that DDR1 is the critical collagen receptor mediating this biology. This is an impactful area and this study provides new insight. Comments below should help improve clarity and impact.

1. In Extended Fig 1. a signature of 6 collagen cleaving MMPs (MMP1, 2, 8, 9, 13, 14) was shown that

is relevant to patient survival. However, more detailed experiments need to be provided regarding the function of the MMPs mentioned. Providing information highlighting which ones are responsible for the cCol or at least a strong discussion on this. What cell types make these MMPs? Macrophages are thought to be the main cells involved in collagen turnover (PMID 29281816), are macrophages critical for cCol activity?

2. Fig. 1a, how was this obtained? If it's through staining, please show representative images.

3. Fig. 1b, 2k, 4h. These experiments seem to have relatively small primary tumors. What are the actual weight of pancreas/tumor tissue in these experiments? IHC for amylase along with H&E of these tumors would be useful.

4. Data that show collagen production levels by WT vs R/R fibroblasts should be provided to exclude the possibility that the effects observed are not due to differential collagen production.

5. Fig. 1i, this is confusing, why is collagen still present in the col IΔ condition? According to extended Fig. 1b, with collagen ablation, there is no iCol or cCol.

6. Fig. 2g, a no col treatment control is needed.

7. Fig. 2k, as mentioned above, many examples of staining here show significant areas of normal pancreas tissue, is this really normal acinar tissue? Moreover, 3/4col, p65, p62, NRF2 staining shown seem to be vessels or lymphoid structures, is this accurate? If so please discuss.

8. Line 222, "These results suggest that NRF2 is critical for cCol I stimulated MP and mitochondrial biogenesis." These two mechanisms should be dissected further. For example, grow NRF2 KD or TFAM KD cancer cells in Col WT and R/R mice.

9. Fig. 3a, the 3/4 Col staining seems to be mostly on cancer cells while full Col is associated with the stroma. Is cleaved Col potentially derived from cancer cells in an autocrine manner?

Author Rebuttals to Initial Comments:

Referees' comments:

Referee #1 (Remarks to the Author):

Hua Su and colleagues describe in this manuscript entitled “Collagen remodeling dictates pancreatic cancer bioenergetics and outcome through DDR1 activation and degradation” a novel role of collagen signaling in pancreatic cancer. The authors describe distinct roles of MMP-cleaved collagen I or full length, intact collagen I in DDR1 biology, activation and down-stream signaling, affecting cancer development. Whereas cleaved Col I promotes PDAC growth through DDR1 signaling (involving NF- κ B/p62 and NRF2) full length, intact Col I induces DDR1 degradation and restrains PDAC growth. Using different mouse models, human tissue and big human cohorts, the authors underline their hypotheses experimentally and show clinical relevance.

Moreover, the included, well characterized patient cohorts to find new stratification criteria for the prognosis of improved median survival are very impressive. This is a highly innovative and clinically relevant study which opens a new field in pancreatic cancer. The study indeed presents a novel pathway in PDAC, opening up several new opportunities for therapy. Conceptually this finding could also be of importance in other cancers with high stromal load.

On the whole the experiments are well executed, well controlled, engaging excellent mouse models and analyses - appropriately chosen and state of the art. This manuscript is very well written and has a red line – enabling the reader to follow the concepts and ideas of the authors in a structure and “good to follow” manner. The most important message of this paper – that Col I remodeling is prognostic for PDAC patient survival – is indeed a big step forward in understanding PDAC pathology in patients – opening novel therapeutic approaches. This referee has a couple of questions/points which might further enhance the clarity of this outstanding manuscript.

Thank you for finding our work of importance and novelty and the many excellent suggestions.

Figure 1:

Figure 1a: these are fascinating data, clearly underlining the central hypothesis of this manuscript. One aspect that would be interesting. Does this the level of cCol I correlate in any way with an altered inflammatory signature, which anyhow in PDAC is special and rather a low profile. E.g. in the context of cCol I and enhanced NF- κ B expression would this lead to any type of alterations?

Thank you for your excellent question. We examined CD45, CD68, CD11c and CD163 expression in cCol I^{high} and cCol I^{low} human PDAC tumors. CD45, CD68 and CD163, but not C68 and CD11c colocalization were lower in cCol I^{low} tumors compared to cCol I^{high} tumors (Extended Data Fig. 7c, Fig. 1a), suggesting that collagen cleavage/remodeling may be associated with tumor

inflammation and altered M1/M2 macrophage populations. This makes good sense because MMP expression is induced by inflammatory stimuli.

Also, we found MMP2 and several inflammatory cytokine/chemokine genes (IL6, TNF, CCL2 and CCL5), MMP8 and TNF, MMP9 and TNF, MMP9 and CCL5, or MMP14 and TNF show strong positive correlations in PDAC patients, based on TCGA database analysis (Fig. 1b). In addition, high IL6 expression correlated with poor survival (Fig. 1c). These results further support the notion that cCol I production correlates with an altered (enhanced) inflammatory signature, as the referee had suspected. (Redacted)

Figure 1b-e: Very impressive in vivo data strongly supporting the hypothesis of the authors.

Thank you for the nice comment.

Figure 1 b, c, d, e: Please add scale bars.

Thank you for the suggestion. We added scale bars to these figures.

Figure 2:

Figure 2b: The single cell analysis data on the expression of DDR1 is highly informative, maybe should be more focused on 1-2 receptors and enlarged. The rest could be shown in the supplemental material.

Thank you for your good suggestion. We kept DDR1 in Fig. 4b and put the other receptors in Extended Data Fig. 4b.

Figure 2c: did the authors also check whether other NF- κ b-related pathways are on besides canonical NF- κ b (nuclear RelA). What about non-canonical NF- κ b signaling, as well as other related pathways? kDa is missing

To answer your question, we examined RelB in KPC960 cells plated on plastic, WT ECM, or R/R ECM. Nuclear RelB amounts did not change in the above cells. Similar results were observed in a mouse hepatocellular carcinoma (HCC) cell line Hepa 1-6 (Fig. II). These results suggest that only canonical, IKK β dependent, but not non-canonical NF- κ B signaling is regulated by Col I remodeling.

Fig. II IB analysis of indicated proteins in KPC960 or Hepa 1-6 cells plated on plastic, WT ECM, or R/R ECM incubated in low glucose (LG) for 24 h.

Figure 2k: Results on Col I controlling MP and mitochondrial biogenesis through the DDR1-NF- κ B-p62-NRF2 cascade are convincing! Just to analyze this in a bit more exact fashion it would be helpful to densitometrically analyze the shown data – at least for the key components DDR1, NF- κ b, p62 – maybe even in a correlation analysis.

Thank you for your comments and nice suggestions. Positive IHC staining of DDR1, p65/RelA, p62, NRF2, NHE1 and SDHB in tumor areas from Col I^{WT} and Col I^{r/r} mice orthotopically or intrasplenically transplanted with KPC960 cells was quantified. Pancreatic and liver tumors from Col I^{r/r} mice showed lower DDR1, p65, p62, NRF2, NHE1 and SDHB compared to tumors from Col I^{WT} mice (Extended Data Fig. 5f, g).

Also here an analysis of possible changes in the most important innate and adaptive immune cells would be very helpful. Any changes in cell survival, or cell death?

According to your good suggestions, we analyzed F4/80, CD45, CD4, CD8 and Ki67 in pancreata isolated from Col I^{WT} and Col I^{r/r} mice in Fig. 4g (original was Fig. 2k). All the above markers, except CD8, were lower in pancreatic tumors from Col I^{r/r} mice compared to Col I^{WT} mice (Extended Data Fig. 10b),

suggesting faster proliferation in Col I^{WT} tumors and different innate and adaptive responses in Col I^{WT} and Col I^{r/r} mice. Further investigation needs to be done to identify the exact cause and impact of these differences, which seem less critical than DDR1 signaling or lack thereof.

Figure 2l: Please add kDa.

Thank you for your suggestion. We added kDa markers.

Figure 2m: Colocalization of GFP-DDR1 and poly-ubiquitin (poly-ub) in GFP-DDR1 expressing 1305 cells cocultured with WT or R/R fibroblasts is very convincing!

Thank you for the nice comment.

Figure 3:

Data in figure 3 underline the clinical relevance of this study. Stains appear very specific and have a low background noise.

Figure 3c: it would be helpful if the authors could show one more blot comparing cCol I low DDR high as well as cCol I low NRF2 hi. In the main Figure.

Thank you for the good suggestion. We analyzed the survival of cCol I^{low}DDR1^{high} vs cCol I^{low}DDR1^{low} and cCol I^{low}NRF2^{high} vs cCol I^{low}NRF2^{low}. Although cCol I^{low}DDR1^{high} or cCol I^{low}NRF2^{high} patients showed worse median survival than those expressing low level of both proteins, there were no significant difference between them due to the small number of patients in each group (Fig. III). We show these data here but prefer not to include them in the manuscript.

Fig. III a, b, Comparisons of overall survival between patients stratified according to cCol I, DDR1 (a) and NRF2 expression (b). Significance was determined by log rank test. * $p < 0.05$.

Data are very convincing demonstrating that the identified pathway indeed is operational in human pathogenesis dictating patient survival.

Figure 4:

Data in Figure 4 are highly convincing, underlining that this pathway is therapeutically targetable.

Importantly, experiments on NF- κ b inhibition (IKK β) or inhibition of mitochondrial protein synthesis causing decreased tumor growth are very impressive.

Minor; Scale bars in all macroscopic images are missing. kDa are missing in Western blots.

Thank you very much for the nice comments and suggestions. We added scale bars and kDa markers to these figures.

On the whole this is an impressive, innovative manuscript with a very focused and well supported message, with extremely high potential for clinical applications to treat or prognostically evaluate PDAC.

Thank you for the enthusiastic and constructive review.

Referee #2 (Remarks to the Author):

Karin and colleagues investigated the role of ECM collagen in the progression of PDAC. PDAC showed a lot of ECM deposition and it may not be only the consequence of the disease. ECM, mainly collagen, in PDAC may also have some functional effects on PDAC growth. Until now, the results are controversial, some reports said collagen promotes, but others reported collagen restricts PDAC growth. This study tried to address this very fundamental but very important question. Therefore, although the study concept is simple, the scientifically, this study is very important. The research team found that MMP-cleaved collagen can promote, but intact collagen has opposite function on PDAC growth. Cleaved collagen activates NF- κ B through the receptor DDR1. Furthermore, they found collagen-induced NF- κ B activation affects the p62-NRF2 signaling to promote PDAC growth by modulating mitochondrial biogenesis. Overall, this study is significant and could lead to a paradigm shift in the field of cancer research.

Thank you for the enthusiastic and constructive review.

This reviewer has several comments that should be addressed.

1. In Fig. 1a, what assay was used for the evaluation of cCol in PDAC patients, either immunoblots or immunostaining? It is unclear from the method description. This reviewer assumes this is immunostaining when looked at Figure 3.

We apologize for the confusion and thank you for raising this point. As you have correctly assumed, we analyzed relative cCol I amount in PDAC specimens by IHC, as shown in Fig. 5a (original was Fig. 3a). We added this information to Fig. 1b (original was Fig. 1a), legend.

2. The authors used MMP1, 2, 8, 9, 13, 14 as a collagen cleavage signature. LOX genes are involved in collagen cross link. Is LOX gene signature also associated with patient outcomes?

To answer your question, we analyzed overall survival of PDAC patients from the TCGA database relative to high and low LOX signature (LOX, LOXL1, LOXL2, LOXL3, LOXL4) and found that high LOX or LOXL2 expression correlated with poor survival (Fig. IVa). Although patients expressed high level of LOXL1 or LOXL3 also showed poor survival, there were no significant differences (Fig. IVa). We also analyzed correlation between the expression of the LOX gene and MMP genes and found strong positive correlations between the LOX gene and several MMP genes (MMP2, 8, 9, 13, 14) (Fig. IVb). These results confirmed that collagen remodeling is a strong prognostic factor.

Fig. IV Correlation between patient survival and LOX gene expression or expression of MMP and LOX genes. a, Overall survival of PDAC patients from TCGA database with high and low LOX signature (Lox, Lox1, Lox2, Lox3, Lox4). Significance was analyzed by log rank test. * $p < 0.05$. **b,** Correlations between the LOX gene and the indicated MMP genes in PDAC specimens from TCGA were analyzed by a Chi-square test. **** $p < 0.0001$.

3. Regarding Extended Data Fig. 1f, the text (line 100) mentioned CCl₄ pretreatment, but the Figure and legend mentioned IKK α KD mice. Please clarify.

We are sorry for the confusion. In Extended Data Fig. 1f, mice were pretreated with CCl₄ and intrasplenically injected with parental or IKK α KD KC6141 cells. In line 100, we compared tumorigenic growth of KPC cells (Fig. 1e, original was Fig. 1d) or KC6141 cells (Extended Data Fig. 1f) in Col I^{WT} and Col I^{r/r} livers: “tumor numbers and size were diminished in Col I^{r/r} livers regardless of CCl₄ pretreatment”.

After we showed that MP-related gene expression is different in tumor cells plated on WT ECM and R/R ECM (Fig. 2c, original was Fig. 1h), we examined this again in Extended Data Fig. 1f, lines 123-126, because IKK α KD KC6141 cells have high MP activity (Su et al., *Cancer Cell*, 2021 PMID: 33740421). We compared tumor growth of parental and IKK α KD KC6141 cells in Col I^{WT} and Col I^{r/r} livers: “Consistent with upregulation of MP genes by WT ECM, IKK α deficient KC6141 cells which have high MP activity formed more tumors than parental KC6141 cells in Col I^{WT} livers, while growing as poorly as parental KC6141 cells in Col I^{r/r} livers”. We hope that it is all clear now.

4. In Fig.1i and Extended Fig. 2f, h, it is unclear why ³H uptake and ATP

production are high in the Col^Δ/Col^{r/r} condition (green bar, vehicle) even no cleaved collagen production (Fig. 1i WB). This reviewer thought cleaved collagen is required for the source of ³H uptake and subsequent ATP production. Both black bar and purple bar have cleaved collagen, but not green bar. To this reviewer and to the authors' conclusion (line 143-144), these data are inconsistent. Please clarify.

Thank you for the excellent question. Initially we thought that Col I needs to be cleaved to be a substrate for MP, but it turned out that cCol I acts as a regulatory molecule. As shown in Fig. 1f (original was Fig. 1e), Fig. 2d (original was Fig. 1i) and Extended Fig. 2f, h, Col I ablated R/R fibroblasts or R/R ECM rescued tumor growth, ³H uptake, ATP, and AA production. These results suggest that in addition to being an MP substrate, cCol I is an important signaling molecule that stimulates PDAC metabolism and energy generation as described in line 143-144 and suggest that removing the inhibitory signal generated by Col I^{r/r} makes the cancer cells able to uptake ³H and get energy from other type of collagens in the Col I^Δ RR ECM. Moreover, these collagens when cleaved by MMPs can also activated DDR1 signaling.

To further answer your question, we plated parental (Paren.) and NRF2(E79Q) activated KPC960 (KPC) cells on ³H labeled WT or R/R ECM +/- EIPA and found while WT ECM increased and R/R ECM decreased ³H uptake in the Paren. cells, inhibition of MP decreased ³H uptake by cells plated on WT ECM. NRF2(E79Q) expressing cells ingested more ³H than the parental cells and were refractory to the inhibitory effect of R/R ECM (Fig. V), suggesting that MP-competent cells can take up ³H from other collagens besides Col I, or from non-collagen ECM components.

Fig. V Parental (Paren.) and NRF2^{E79Q} (E79Q) KPC cells grown on ³H-proline-labeled WT, or R/R ECM were incubated in LG medium +/- EIPA (10.5 μM) for 24 h. ³H uptake was measured by liquid scintillation, normalized to cell number, and presented as ³H CPM relative to vehicle treated WT ECM plated Paren. KPC cells. Mean ± SEM (n=3). Statistical significance was determined by a two-tailed t test. ***p < 0.001, ****p < 0.0001.

Moreover, we had shown that overexpression of NRF2(E79Q) in DDR1 ablated-KPC cells reversed the inhibitory effects of R/R ECM or DDR1 ablation on TMR-DEX uptake but not on 3/4 Col I fragment uptake (Extended Data Fig. 4e). In addition, we mixed R/R with WT (R:W) or Col I^Δ R/R (R:KO) fibroblasts to generate ECM with different relative amounts of iCol I and cCol I, as confirmed by blotting with isoform specific antibodies. KPC cells were plated

on the different ECM preparations. Non degradable Col I at 6:4 (R:W) or 4:6 (R:KO) ratios and above inhibited MP and reduced mitochondria numbers and nuclear NRF2 (Extended Data Fig. 3a, b). These results suggest that iCol I inhibits MP and mitochondrial biogenesis, which are stimulated by different DDR1 activating cleaved collagens, and not only cCol I. Indeed, pancreatic cancer cells grown on R/R Col I^Δ ECM have higher MP activity and higher mitochondrial content due to elevated DDR1-NRF2 signaling compared to cells plated on R/R ECM, while DDR1 ablation reversed this (Extended Data Fig. 6a-c). These results confirm that Col I is not only an MP substrate used as an energy source but is also an important signaling molecule that regulates tumor metabolism. As long as Col I is cleaved it activates DDR1, which can also be activated by other cleaved collagens produced by Col I^Δ fibroblasts. These results also explain why NRF2(E79Q), which acts downstream to DDR1, rescues tumor growth regardless of the Col I status (Fig. 6e-g).

5. In primary PDAC, DDR1 expression seems low in primary PDAC (Fig. 2b). In contrast, ITGB1 expression is high in both primary and metastatic PDAC (Extended Fig 4b). Because the functional test showed DDR1 is more important than ITGB1, these scRNA-seq data seems not nice. Because the sample number is not many, either removing, adding more sample number, or other staining approaches (immunostaining or RNA scope) is recommended.

Previously, we used a batch-corrected and merged PDAC and 5 different PDAC metastases datasets from Lee et al 2021 (PMID: 34426439) for analysis. To improve clustering accuracy, we have analyzed PDAC and liver met data separately, as shown in Fig. 4b (original was Fig. 2b) and Extended Fig. 4b. *ITGB1* mRNA expression was still higher in this improved analysis than *DDR1* mRNA.

In the directed CRISPR-CAS9 screen shown in Extended Data Fig. 4a, we identified the collagen receptor that senses the stimulatory effects of cCol I on MP and mitochondrial content in PDAC cells (Fig. 4a). This experiment unequivocally established the role of DDR1 and showed that ITGB1 is not important in this context, although it probably has other functions that we have not assayed for. We then used single cell analysis to compare the expression of different collagen receptors in primary and liver metastatic PDAC. Importantly, DDR1 was expressed mainly by the tumor epithelial cells in both cases. As ranked by mRNA amounts, ITGB1 was higher than DDR1 in both primary and metastatic PDAC, but it also had a very broad distribution, being expressed in myeloid, T cells, NK cells, B cells, and fibroblasts in addition to tumor epithelial cells. We agree that the results may seem not nice, but these are the results and we think that ITGB1 should be included rather than excluded.

6. Line 196-197, if the authors want to claim “....., with other collagens in Col delta fibroblasts acting as ligands”. It could be required to use collagenase to

degrade all collagens, and then the study needs to show that effect can be disappeared. Depending on the result, there is a possibility that DDR1 ligands, other than collagens, might stimulate the signaling though the impact of this study does not change.

Thank you for the excellent question and suggestion. We used collagenase (CLG) from *Clostridium histolyticum* which hydrolyzes native collagens into a mixture of small peptides (Zhang et al., Appl Environ Microbiol., 2015. PMID: 26150451). KPC cells were plated on plastic, or ECM produced by CLG incubated or not-incubated WT and R/R fibroblasts and cultured in LG medium +/- CLG for 24 h. ECM from fibroblasts treated with CLG no longer activated DDR1-NRF2 signaling, MP activity and mitochondrial biogenesis (Fig. VI). Similarly, CLG treatment inhibited DDR1-NRF2 signaling in KPC cells plated on ECM from Col I^Δ WT or Col I^Δ RR fibroblasts (Extended Data Fig. 6d). Thus, in the absence of Col I, other collagens serve as DDR1 ligands. These results are entirely consistent with those of the pan-MMPi experiment, which prevents cleavage of all collagens (Extended Data Fig. 6f, g)

Fig. VI Collagen removal blocks DDR1-NRF2 mediated MP activation and mitochondrial biogenesis. **a**, IB analysis of indicated proteins in 1305 cells plated on plastic or ECM produced by CLG (50 μ g/ml) incubated or not incubated WT or RR fibroblasts and cultured in LG medium +/- CLG for 24 h. **b**, Imaging and quantification of MP and mitochondria in 1305 cells plated on ECM produced by CLG treated or untreated WT or RR fibroblasts and incubated in LG medium +/- CLG for 24 h. Mean \pm SEM (n=6). Statistical significance was determined by a two-tailed t test. **** p < 0.0001.

Referee #3 (Remarks to the Author):

Many cancer types are characterized by a fibrotic phenotype with abundant collagen deposition. However, in the field there is inconsistent data regarding the function of collagen in tumor progression. The current study demonstrates that a tumor-promoting function is mediated by cleaved collagen (cCol). cCol enhances bioenergetics and macropinocytosis in pancreatic cancer through DDR1, a RTK that binds triple helical collagen. In contrast the authors show that intact collagen (iCol) promotes DDR1 degradation and has an inhibitory effect on cancer cells. This study provides some clarity on the function and importance of signaling initiated by cleaved collagen in pancreatic cancer progression. Further the study demonstrates that DDR1 is the critical collagen receptor mediating this biology. This is an impactful area and this study provides new insight. Comments below should help improve clarity and impact.

Thank you for finding our study important and impactful, as well as the excellent suggestions.

1. In Extended Fig 1. a signature of 6 collagen cleaving MMPs (MMP1, 2, 8, 9, 13, 14) was shown that is relevant to patient survival. However, more detailed experiments need to be provided regarding the function of the MMPs mentioned. Providing information highlighting which ones are responsible for the cCol or at least a strong discussion on this. What cell types make these MMPs? Macrophages are thought to be the main cells involved in collagen turnover (PMID 29281816), are macrophages critical for cCol activity?

Thank you for the good questions and suggestions. All of the 6 MMPs we have examined (MMP1, 2, 8, 9, 13 14) can degrade Col I (Pittayapruek et al., *Int J Mol Sci.*, 2016 PMID: 27271600; Page-McCaw et al., *Nat Rev Mol Cell Biol.*, 2007 PMID: 17318226). Col I is the most abundant fibrillar collagen present in the PDAC ECM (Tian et al., *Proc Natl Acad Sci USA*, 2019, PMID: 31484774). Therefore, we choose them to generate an MMP signature and investigated their relevance to patient survival.

To investigate which cell types are responsible for MMP production, scRNA analysis of *MMP* mRNAs in human PDAC were conducted. Consistent with previous studies (Pittayapruek et al., *Int J Mol Sci.*, 2016 PMID: 27271600), MMPs were highly expressed in multiple cell types, epithelial-tumor, myeloid and fibroblasts (Extended data Fig. 1b).

As you have mentioned (PMID: 29281816), we agree that macrophages are involved in collagen turnover because in addition to MMP production, macrophages possess constitutive macropinocytosis (Canton, *Front Immunol.*, 2018 PMID: 30333835). In addition, we found cCol I amounts correlate with an inflammatory signature in human and mouse PDAC (Extended data Fig. 7c, 10b) and upregulation of M2 macrophages in human PDAC (Extended data Fig. 7c). Macrophages and especially M2-like macrophages highly expressed

multiple MMPs indicated by scRNA analysis of human PDAC (Extended Data Fig. 1b). We therefore fully agree that macrophages play an important role in cCol I generation. Moreover, we had shown that ECM from fibroblasts treated with the FDA approved pan-MMPi Ilomastat act like R/R ECM, inhibiting DDR1-NRF2 signaling, MP activity and mitochondrial biogenesis, all of which were reversed by activated NRF2(E79Q) overexpression (Extended Data Fig. 6f, g). These results suggest that MMP-mediated collagen cleavage is a key regulator of tumor metabolism. How Col I remodeling regulates macrophage behavior and how macrophages fully regulate tumor growth need to be addressed in future studies.

2. Fig. 1a, how was this obtained? If it's through staining, please show representative images.

We are sorry for the incomplete description. These data were obtained by analyzing relative cCol I amount by the IHC shown in Fig. 5a (original was Fig. 3a). We added this information to Fig. 1b (original was Fig. 1a) legend.

3. Fig. 1b, 2k, 4h. These experiments seem to have relatively small primary tumors. What are the actual weight of pancreas/tumor tissue in these experiments? IHC for amylase along with H&E of these tumors would be useful.

Thank you for the excellent question and suggestion. In Fig. 4g (original was Fig. 2k), the pancreata were derived from untreated mice used in Fig. 1c (original was Fig. 1b). We had shown H&E, amylase, CK19 and SOX9 staining (Extended Data Fig. 1c, d) of these pancreata in Fig. 1c. According to your suggestions, we also did H&E, amylase and CK19 staining (Extended Data Fig. 10c) of the pancreata in Fig. 6g (original was Fig. 4h) and quantified tumor areas of above pancreata according to amylase and CK19 staining (Extended Data Fig. 1d, 10c). Large areas of the pancreas retained observed normal tissue in CAE-untreated Col I^{WT} and Col I^{r/r} mice or CAE-treated Col I^{r/r} mice orthotopically transplanted with Paren. or DDR1 ablated KPC cells.

4. Data that show collagen production levels by WT vs R/R fibroblasts should be provided to exclude the possibility that the effects observed are not due to differential collagen production.

We fully understand your concern. To exclude the possibility that the observed effects are due to differential collagen production by WT and R/R fibroblasts, we detected Col I and total collagen from fibroblasts treated +/- the FDA-approved MMPi Ilomastat (Iloma.) by IB and Sirius red (SR) staining. WT and R/R fibroblasts produced equal amounts of Col I after Iloma. treatment and equal amounts of total collagen regardless of Iloma. treatment (Fig. VII).

In addition, ECM from fibroblasts treated with Iloma. behaved like R/R ECM, inhibiting MP activity, mitochondrial biogenesis, and DDR1-NRF2 signaling, all of which were reversed by activated NRF2(E79Q) overexpression (Extended Data Fig. 6f, g). Col I ablation in R/R fibroblast wiped out the inhibitory effect of

R/R ECM on DDR1-NRF2 signaling, MP activity and mitochondrial biogenesis (Extended Data Fig. 6a, b). Moreover, inhibition of DDR1 repressed MP and mitochondrial biogenesis in KPC cells plated on WT ECM or tumor growth in Col I^{WT} pancreas, while NRF2(E79Q) overexpression restored them in KPC cells plated on R/R ECM or grown in Col I^{r/r} pancreas (Figs. 4a, d and 6g). These results suggest that the Col I state and its differential effect on DDR1-NRF2 signaling, but not its amount, controls PDAC metabolism.

Fig. VII a, Imaging and quantification of Sirius Red (SR) staining of ECM produced by WT or RR fibroblasts +/- Iloma. Treatment. SR density was measured on a microplate reader at 540 nm. Mean \pm SEM (n=5). Statistical significance was determined by a two-tailed t test. **b**, IB analysis of indicated proteins in the ECM produced by above fibroblasts.

5. Fig. 1i, this is confusing, why is collagen still present in the Col I^A condition? According to extended Fig. 1b, with collagen ablation, there is no iCol or cCol.

We are sorry for the confusion. In Fig. 2d (original was Fig. 1i), we re-expressed Col I^{WT} in Col I^A R/R fibroblasts (the fourth sample), which was labeled with “+ Col I^{WT}”. Therefore, Col I is still present in this sample. We clarified this in the legend.

6. Fig. 2g, a no col treatment control is needed.

According to your request, we added no Col treatment control in Fig. 4f (original was Fig. 2g).

7. Fig. 2k, as mentioned above, many examples of staining here show significant areas of normal pancreas tissue, is this really normal acinar tissue? Moreover, 3/4col, p65, p62, NRF2 staining shown seem to be vessels or lymphoid structures, is this accurate? If so please discuss.

As mentioned above, these pancreata derived from Col I^{WT} and Col I^{r/r} mice 4 weeks after orthotopic KPC cell transplantation without CAE pretreatment, shown in Fig. 1c, Extended Data Fig. 1c, d. Although Col I^{r/r} mice poorly support primary pancreatic tumor growth, large areas of pancreas remained normal tissue in CAE-untreated Col I^{WT} and Col I^{r/r} mice as shown the H&E staining and IHC analysis of amylase, CK19 and SOX9 of these pancreata (Extended Data Fig. 1c, d), because even in Col I^{WT} mice the tumors occupy only a part of the pancreas. We repeated these experiments by H&E staining and IHC analysis of amylase and CK19 and confirmed that the normal appearing pancreas is really normal acinar tissue (Fig. VIIIa).

As shown in Fig. 4g (original was Fig. 2k), pancreatic tumors from Col I^{r/r} mice showed more extensive iCol I expression but no cCol I and lower DDR1, p65, p62, NRF2, NHE1 and SDHB (mitochondrial marker) compared to tumors from Col I^{WT} mice. 3/4 Col I (cCol I), p65, p62 and NRF2 stained areas were not blood vessels or lymphoid structures as indicated by CK19 IHC of serial sections from the pancreata shown in Fig. 4g (Fig. VIIIb).

Please note that the 3/4 Col I and NRF2 stainings were performed on serial sections of tumor bearing Col I^{r/r} pancreata shown in Fig. 4g, with CK19 staining shown below. Also please note that we changed the magnification of the 3/4 Col I stained area according to the CK19 positive area in Fig. 4g. Although the CK19 stained area (red dashed area) paralleling the 3/4 Col I stained section of Col I^{WT} pancreata was not 100% positive, this area was surrounded by stoma cells which secreted cleavable Col I ingested by cancer cells.

Fig. VIII a, H&E staining and IHC analysis of amylase and CK19 in pancreatic sections from tumor bearing Col I^{WT} and Col I^{r/r} mice in Fig. 4g. **b**, IHC analysis of CK19 in serial sections corresponding to those shown in Fig. 4g. The antibody used for staining the parallel sections in Fig. 4g are indicated in the parentheses. Please note we used same CK19 image at the left bottom in a and b for answering the referee's question.

8. Line 222, “These results suggest that NRF2 is critical for cCol I stimulated MP and mitochondrial biogenesis.” These two mechanisms should be dissected further. For example, grow NRF2 KD or TFAM KD cancer cells in Col WT and R/R mice.

This is a good suggestion. In previous studies (Su et al., Cancer Cell, PMID: 33740421) we showed that NRF2 is the transcriptional activator of the MP program. Here we found cCol I and DDR1 stimulate MP by activating NRF2 (Fig. 4c, d, f). NRF2 stimulates mitochondrial biogenesis by inducing TFAM (Extended Data Fig. 6j-m). According to your suggestion, NRF2^{KD} and TFAM^{KD}

KPC cells were generated and orthotopically transplanted into Col I^{WT} or Col I^{r/r} pancreata. NRF2 or TFAM ablation inhibited tumor growth in the Col I^{WT} pancreas but did not reduce it further in the Col I^{r/r} pancreas (Extended Data Fig. 11c, d).

9. Fig. 3a, the 3/4 Col staining seems to be mostly on cancer cells while full Col is associated with the stroma. Is cleaved Col potentially derived from cancer cells in an autocrine manner?

MP is an evolutionarily conserved form of endocytosis that mediates non-selective uptake of extracellular fluid and the solutes contained therein. Therefore, cancer cells can only ingest $\frac{3}{4}$ (or $\frac{1}{4}$) Col I (soluble) by MP but not full-length Col I (insoluble). We have shown that cancer cells plated on WT ECM can ingest $\frac{3}{4}$ Col I through MP indicated by co-localization of $\frac{3}{4}$ Col I and TMR-DEX (label macropinosome), which was inhibited by R/R ECM (Extended Data Fig. 4e). Most of the cleavable Col I cleaved by MMPs is generated by fibroblasts and is present in the extracellular milieu. If cleaved Col I was derived from the small amounts of collagen expressed by the cancer cells, $\frac{3}{4}$ Col I in KPC tumors isolated from Col I^{r/r} mice should have been detected. However, this was only observed in KPC tumors isolated from Col I^{WT} mice (Fig. 4g and Extended Data Fig. 5g). Therefore, the majority of $\frac{3}{4}$ Col I staining on cancer cells is likely derived from the tumor microenvironment.

Reviewer Reports on the First Revision:

Referees' comments:

Referee #1 (Remarks to the Author):

The authors have answered all my questions experimentally and have clarified all open points. This will become an impactful paper in the PDAC field and will ignite more research in PDAC in this direction.

In my point of view this paper is now deemed appropriate for publication in Nature.

Referee #2 (Remarks to the Author):

The authors properly addressed all comments/concerns raised by this reviewer. No additional comments.

Referee #3 (Remarks to the Author):

The authors have done a commendable job in addressing the queries of the original review.

Strong study, no further comments for the authors that need to be addressed. Remaining questions revolve around the activity of proteases on insoluble vs soluble collagens and whether other collagen proteases (e.g., cathepsins) are relevant, these are not central or critical to the current manuscript.

The authors could also put their study in context of prior work that has examined the importance of collagen-DDR1 signaling in pancreatic cancer. For example, endogenous inhibitors of DDR signaling are present in the tumor microenvironment. One of these is the matricellular protein SPARC. SPARC binds to collagen I at the GVMGFO site, which is the same site on collagen that binds DDR1 (PMID: 19011090, 20004161). SPARC competes with DDR1/2 for binding to collagen (PMID: 21666116, 24346431). The absence of SPARC leads to increased DDR1 activation in pancreatic cancer (PMID: 24346431) and more aggressive growth of pancreatic tumors (PMID: 20007485). Additionally, recent studies have examined the impact of collagen-induced DDR1 activity in tumor cells on the biology of pancreatic tumor progression (PMID: 34237033). These studies in aggregate provide a contextual backdrop for the current manuscript and highlight the importance of mechanistic insight provided by the manuscript from Su et al.

Minor comment

Abstract Line 51 add 'to' after desmoplastic cancers

Author Rebuttals to First Revision:

Referees' comments:

Referee #3 (Remarks to the Author):

The authors have done a commendable job in addressing the queries of the original review.

Strong study, no further comments for the authors that need to be addressed. Remaining questions revolve around the activity of proteases on insoluble vs soluble collagens and whether other collagen proteases (e.g., cathepsins) are relevant, these are not central or critical to the current manuscript.

The authors could also put their study in context of prior work that has examined the importance of collagen-DDR1 signaling in pancreatic cancer. For example, endogenous inhibitors of DDR signaling are present in the tumor microenvironment. One of these is the matricellular protein SPARC. SPARC binds to collagen I at the GVMGFO site, which is the same site on collagen that binds DDR1 (PMID: 19011090, 20004161). SPARC competes with DDR1/2 for binding to collagen (PMID: 21666116, 24346431). The absence of SPARC leads to increased DDR1 activation in pancreatic cancer (PMID: 24346431) and more aggressive growth of pancreatic tumors (PMID: 20007485). Additionally, recent studies have examined the impact of collagen-induced DDR1 activity in tumor cells on the biology of pancreatic tumor progression (PMID: 34237033). These studies in aggregate provide a contextual backdrop for the current manuscript and highlight the importance of mechanistic insight provided by the manuscript from Su et al.

Minor comment

Abstract Line 51 add 'to' after desmoplastic cancers

Thank you for your nice comments. We added 'to' after 'desmoplastic cancers' in Abstract and discussed the potential role of SPARC.